# The budding yeast Centromere DNA Element II wraps a stable Cse4 hemisome in either orientation in vivo

Steven Henikoff[1,2]*, Srinivas Ramachandran[1,2], Kristina Krassovsky[1,3], Terri D Bryson[1,2], Christine A Codomo[1], Kristin Brogaard[4], Jonathan Widom[4†], Ji-Ping Wang[5], Jorja G Henikoff[1]

[1]Basic Sciences Division, Fred Hutchinson Cancer Research Center, Seattle, United States; [2]Howard Hughes Medical Institute, Fred Hutchinson Cancer Research Center, Seattle, United States; [3]Molecular and Cellular Biology Program, University of Washington, Seattle, United States; [4]Department of Molecular Biosciences, Northwestern University, Evanston, United States; [5]Department of Statistics, Northwestern University, Evanston, United States

**Abstract** In budding yeast, a single cenH3 (Cse4) nucleosome occupies the ~120-bp functional centromere, however conflicting structural models for the particle have been proposed. To resolve this controversy, we have applied H4S47C-anchored cleavage mapping, which reveals the precise position of histone H4 in every nucleosome in the genome. We find that cleavage patterns at centromeres are unique within the genome and are incompatible with symmetrical structures, including octameric nucleosomes and (Cse4/H4)$_2$ tetrasomes. Centromere cleavage patterns are compatible with a precisely positioned core structure, one in which each of the 16 yeast centromeres is occupied by oppositely oriented Cse4/H4/H2A/H2B hemisomes in two rotational phases within the population. Centromere-specific hemisomes are also inferred from distances observed between closely-spaced H4 cleavages, as predicted from structural modeling. Our results indicate that the orientation and rotational position of the stable hemisome at each yeast centromere is not specified by the functional centromere sequence.

*For correspondence: steveh@fhcrc.org

†Deceased

Competing interests: The authors declare that no competing interests exist.

## Introduction

Centromeres are the genetic loci that organize the proteinaceous kinetochore, which attaches to spindle microtubules to pull the chromosomes to the poles in both mitosis and meiosis. There is general agreement in the centromere field that the central determinant of centromere identity and propagation is the special centromeric nucleosome containing the cenH3 (CENP-A in mammals and Cse4 in budding yeast) histone variant (*Quenet and Dalal, 2012*). CenH3 nucleosomes have been shown to occupy the centromeres of nearly all eukaryotes studied, and to be necessary for kinetochore formation. Despite the central importance of this nucleosome, its composition and structure have been the subject of controversy. In vitro and in vivo studies have led to proposals for several mutually exclusive models, including conventional octameric (cenH3/H4/H2B/H2A)$_2$ nucleosomes ('octasomes') (*Camahort et al., 2009*), cenH3/H4/H2B/H2A half-nucleosomes ('hemisomes') (*Dalal et al., 2007*), (cenH3/H4)$_2$ tetrasomes (*Aravamudhan et al., 2013*), mixed (cenH3/H3/H4$_2$/H2B$_2$/H2A$_2$) octasomes (*Lochmann and Ivanov, 2012*) and (cenH3/H4/Scm3)$_2$ hexasomes (*Mizuguchi et al., 2007*), where Scm3 is a cenH3-specific histone chaperone.

Evidence for each of these conflicting models has been presented for budding yeast, where the centromere is genetically defined by an ~120-bp functional sequence on each of the 16 chromosomes.

**eLife digest** DNA is tightly packaged in cells for a variety of reasons—to allow it to fit inside the nucleus, to protect it from damage, and to help control the production of proteins from genes. The basic unit of packaged DNA is called a nucleosome, which consists of DNA wrapped around a structure formed by two pairs of four different proteins.

These proteins, which are called histones, have a role that extends beyond providing structural support for DNA. When cells divide, for example, pairs of 'sister chromosomes' are pulled apart to ensure that the two daughter cells both have the same chromosomes as the original cell. The sister chromosomes are pulled apart from a single position called a centromere, and the nucleosomes at this position contain a histone that is different from the histones found everywhere else in the cell. However, until recently it was not clear if the nucleosomes that contained these special cenH3 histones had the same structure as other nucleosomes.

Now Henikoff et al. have used a method called H4S47C-anchored cleavage mapping to study every nucleosome in the genome of the yeast *S. cerevisiae*. This mapping technique uses DNA sequencing to measure the precise distances between fixed points on the DNA in the nucleosome. Knowing these distances tells researchers a great deal about the number and position of the histones within each nucleosome in the genome.

Using this approach, Henikoff et al. found that nucleosomes at centromeres are different from other nucleosomes in histone number and arrangement. In particular, the nucleosome at each yeast centromere contains only one each of the four different histones in an asymmetrical orientation, in contrast to all other yeast nucleosomes, which contain two sets of four histones in a symmetrical arrangement. Furthermore, each nucleosome at a centromere can adopt one of two orientations: these orientations are mirror images of each other, and they occur with equal probability. It should also be possible to use the mapping technique developed by Henikoff et al. to study the larger and more complex centromeres found in other organisms, including humans.

The functional centromere has a tripartite organization: the 8 bp CDEI sequence is a binding site for the Cbf1 protein, the Cse4 nucleosome maps to the 78–86 bp CDEII sequence, and the 26 bp CDEIII sequence is a binding site for the Cbf3 complex (http://www.yeastgenome.org). Our previous native chromatin immunoprecipitation (ChIP) study has resolved all three particles at base-pair resolution, confirming that the Cse4-containing nucleosome is confined to CDEII (*Krassovsky et al., 2012*). The implied single wrap of DNA around the Cse4-containing histone core is consistent with previous evidence that the Cse4 nucleosome wraps DNA in a right-handed orientation in vivo (*Furuyama and Henikoff, 2009*; *Huang et al., 2011*), opposite to the left-handed wrap of conventional nucleosomes (*Tachiwana et al., 2011*). However, conflicting structural interpretations have continued to appear, with some authors arguing for partially unwrapped octasomes (*Dunleavy et al., 2013*; *Hasson et al., 2013*; *Miell et al., 2013*; *Padeganeh et al., 2013*), others for cenH3/H4 octasomes (*Lochmann and Ivanov, 2012*), others for tetrasomes (*Aravamudhan et al., 2013*), and others for hemisomes throughout the cell cycle but octasomes at anaphase (*Shivaraju et al., 2012*). Although there have been suggestions of more than one Cse4 nucleosome per budding yeast centromere based on fluorescent microscopy (*Coffman et al., 2011*; *Lawrimore et al., 2011*), more recent evidence confirms that there is only one particle per centromere (*Henikoff and Henikoff, 2012*; *Shivaraju et al., 2012*; *Haase et al., 2013*). As budding yeast is the only model organism where there is a 1:1 relationship between the cenH3 nucleosome and the microtubule attachment site (*Furuyama and Biggins, 2007*), any conclusion concerning its composition and structure has an unequivocal functional interpretation.

To definitively settle the controversy over budding yeast centromeric nucleosome composition and structure, we have turned to an in vivo mapping method that individually characterizes every nucleosome in the genome. H4S47C-anchored cleavage mapping determines the precise position and orientation of all histone H4 molecules in an unbiased manner (*Brogaard et al., 2012a*). By mapping the obligate H4 partner of Cse4, we avoid potential complications arising from the need for antibodies, tags, nucleases or fluorescence. As H4 is the obligate partner of every H3 in the genome, this method

provides ~75,000 positive control H3 nucleosomes to compare against the 16 Cse4 nucleosomes at centromeres. There are important advantages to this approach to mapping yeast centromeres over previous mapping methods, such as Micrococcal Nuclease with sequencing (MNase-seq) and native ChIP. Unlike MNase, an endo-exonuclease that preferentially cleaves linkers between nucleosomes and so provides a map of regions protected from cleavage, H4S47C-anchored cleavage mapping determines base-pair positions *within* the particle. This means that mapping is not complicated by exonucleolytic 'nibbling' and internal cleavages that can lead to uncertainty as to the true size of a particle. In addition, CDEIIs are >90% A+T, and MNase prefers AT-rich regions, resulting in preferential exonucleolytic digestion and loss of centromeric DNA, whereas H4S47C-anchored cleavage is exclusively endonucleolytic and has no sequence bias. When combined with paired-end deep sequencing, H4S47C-anchored cleavage mapping provides precise center-to-center distances between adjacent particles that can be used to infer interactions between neighboring nucleosomes and to probe higher-order structural properties without chromatin solubilization. Most importantly for the problem being addressed in this study, H4S47C-anchored cleavage mapping predicts very different cleavage patterns for octasomes and tetrasomes, which are symmetrical, than for hemisomes, which are asymmetrical.

Using H4S47C-anchored cleavage mapping we show that both the overall pattern and the distances between cleavages are incompatible with all proposed octasome or tetrasome particles. Rather, the pattern and cleavage distances indicate mutually exclusive occupancy of CDEII by oppositely oriented hemisomes in two rotational phases at similar frequencies. Our findings reveal surprising flexibility in orientation and phasing for a nucleosome particle that is tightly confined within an asymmetric ~80-bp DNA loop.

## Results

### A molecular dynamics model for H4S47C-anchored hydroxy radical cleavage

In the original description of H4S47C-anchored cleavage mapping, nucleosome centers were determined as clusters of cleavages around the dyad axes of highly occupied and phased nucleosomes throughout the budding yeast genome (*Brogaard et al., 2012b*). It appeared that cleavages at centromeres were different from cleavages at other nucleosomes, although the empirical model used to interpret cleavage patterns did not permit further inferences concerning the structure and composition of the particle over centromeric DNA. Specifically, cleavages at the two closely spaced H4S47C residues can potentially result in <10-bp DNA duplexes that cannot be uniquely mapped in the genome, making the empirical approach unsuitable for discriminating cleavage patterns generated by single H4S47C residues in Cse4 hemisomes from those generated by two H4S47C residues in a canonical nucleosome. Therefore, we applied molecular dynamics simulations based on the high-resolution structure of the nucleosome core particle (*Davey et al., 2002*) with a single phenanthroline group attached by derivatizing the cysteine sulfide with N(1,10-phenanthroline-5-yl)iodoacetamide (*Figure 1A*). The phenanthroline moiety chelates a copper ion, and when it contacts the C-1 hydrogen (C1H) of deoxyribose through a hydroxyl radical, initiates a series of elimination reactions that result in strand cleavages, releasing the deoxyribonucleotide and leaving 5'- and 3'- phosphate ends (*Sutton et al., 1993*). The simulations sampled possible conformations of a single H4S47C-phenanthroline-Cu$^+$ in the context of the nucleosome, while the DNA and the rest of the protein were held static (*Video 1*). From these simulations, we observed that the copper can be within 4 Å of the C1H atoms at −2, −3, and −4 on the Watson strand and −5, −6 on the Crick strand with respect to the nucleosome dyad axis (*Figure 1B*). Contacts on both strands imply that a single H4S47C-phenanthroline-Cu$^+$ can catalyze double-strand cleavages.

In an octasome, H4S47C-phenanthroline-Cu$^+$ contacts made by the first H4 correspond to contacts at +5 and +6 on the Watson strand and +2, +3 and +4 on the Crick strand made by the second H4. We interpret the contacts made by both H4s as predictive of single-strand cleavages creating 5' and 3' ends on either side of the deoxyribonucleotide under attack (e.g., *Figure 1C*). Importantly, these predicted cleavages on both sides of the dyad axis are among those deduced by Brogaard et al. from their empirical in vivo data (*Brogaard et al., 2012b*), and confirmed by cleavage mapping of reconstituted octasomes (*Figure 1—figure supplement 1*).

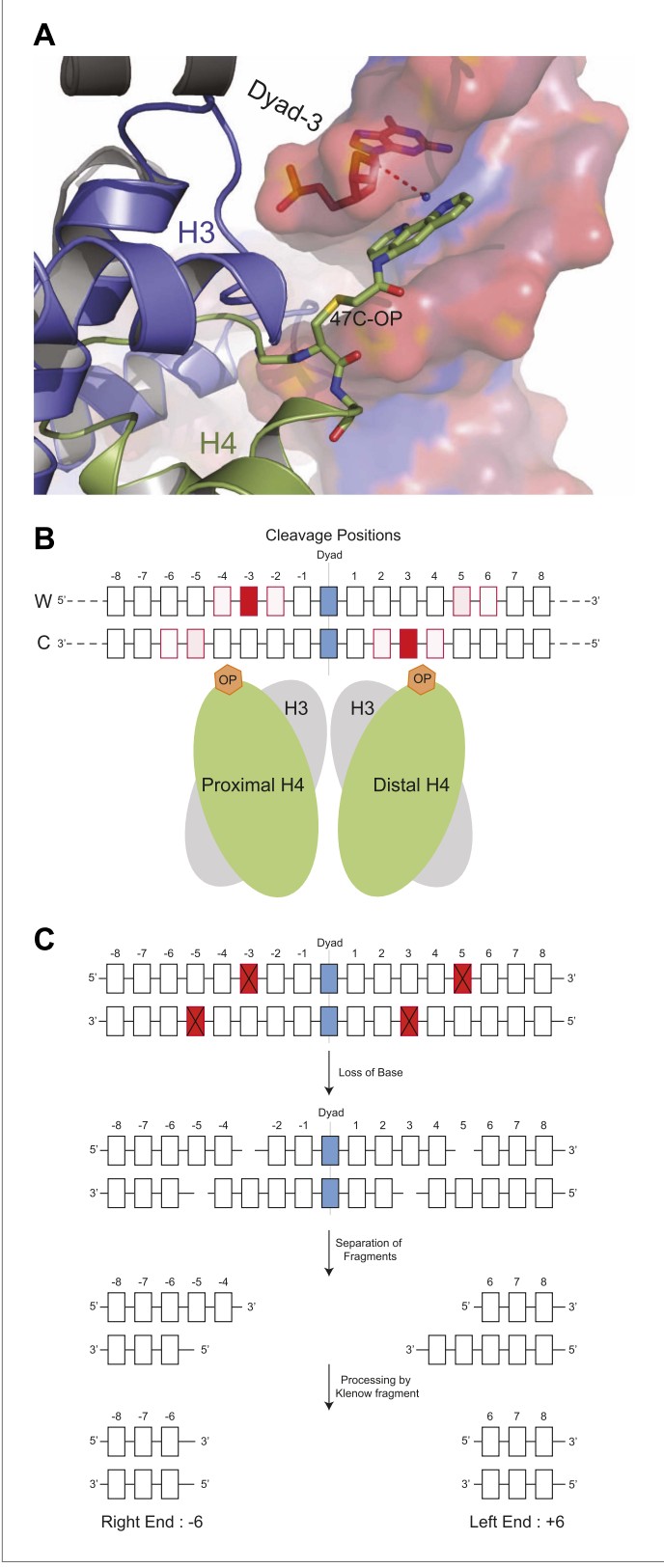

**Figure 1**. Structural model for DNA cleavage by H4S47C-phenanthroline-Cu$^+$. (**A**) Snapshot from a molecular dynamics simulation showing a copper ion (blue dot) bound to phenanthroline within 4 Å (dotted line) of the C1H atom of the deoxyribonucleotide at the Dyad-3 position (red). Phenanthroline (shown in stick representation) docks

*Figure 1. Continued on next page*

*Figure 1. Continued*

in the minor groove of the nucleosomal DNA (shown in surface representation). (**B**) Cleavage positions with respect to the dyad axis on the Watson (W) and Crick (C) strands. The degree of red shading corresponds to the probability of predicted contact. (**C**) H4S47C-anchored cleavage and processing. The steps involved in H4S47C-anchored cleavage mapping are illustrated for an instance in which H4S47C-Phenanthroline-Cu cleavages occur on both sides of the dyad. The cleaved positions are indicated as red squares marked with crosses. In this instance, the W-C distance would be +12. Note that that right end of a fragment marks the cleavage position on the Crick strand, and the left end marks the cleavage position on the Watson strand. See also *Figure 1—figure supplement 1*.

The following figure supplements are available for figure 1:

**Figure supplement 1**. Determination of cleavage positions by in vitro cleavage mapping.

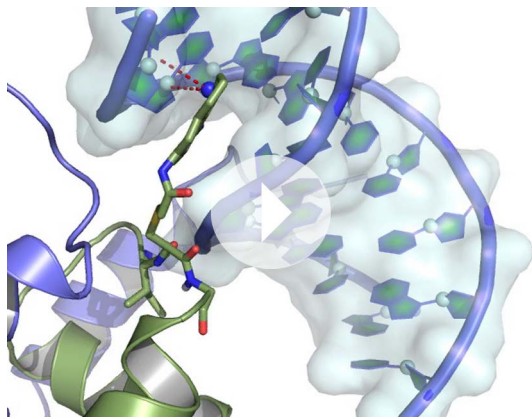

**Video 1**. Molecular dynamics simulation of DNA contacts by a copper ion chelated to H4S47C-anchored 1, 10-orthophenanthroline. See the legend to *Figure 1A*.

## A unique pattern of paired cleavages characterizes all yeast centromeres

Previous high-resolution MNase and native-ChIP mapping of *Saccharomyces cerevisiae* centromeres revealed that the distance between the midpoint of the Cse4 nucleosome occupying CDEII to the midpoint of immediately flanking nucleosomes is ~200–250 bp (*Cole et al., 2011*; *Krassovsky et al., 2012*). In the original study using H4S47C-anchored cleavage, fragments were size-selected to be mostly 125–200 bp, which resulted in very low coverage of individual centromeres (*Brogaard et al., 2012b*). To avoid such size-related biases, we followed essentially the same protocol in generating cleavage products, but applied a modified Solexa paired-end DNA sequence library preparation protocol that does not include a size-selection step (*Henikoff et al., 2011*). In both biological and technical replicates using several variations in the protocol ('Materials and methods'), we obtained virtually identical fragment length distributions (*Figure 2A*), indicating a high degree of repeatability and robustness in the basic cleavage mapping protocol. Centromere function is normal in the H4S47C mutant cell line, as we observed virtually identical doubling times of 90 min and a CEN plasmid retention rate of 99% per generation for the H4S47C strain (*Figure 2—figure supplement 1*).

We mapped paired-end reads to the budding yeast genome, and except as noted, pooled all 172,272,295 mapped fragments for the analyses below. As nucleosome center-to-center distances should be ≥147 bp, we excluded fragments <147 bp, which resulted in a cleavage map (*Figure 2B,C*) that is very similar to that described previously (*Brogaard et al., 2012b*). This confirms that cleavage clusters correspond to nucleosomes throughout the genome. However, the cleavage pattern over centromeres is different from the large majority of cleavages on chromosome arms. As previously noted for all 16 aligned centromeres (*Brogaard et al., 2012b*), individual centromeres display two cleavage clusters spaced ~40 bp apart centered over the middle of each CDEII (*Figure 2D,F,H,J*).

MNase-seq and Cse4 native ChIP studies have shown that the Cse4 nucleosome is almost perfectly aligned over CDEII of each functional centromere, and we wondered whether the double peak pattern of cleavage clusters over each centromere is seen at other highly occupied and phased nucleosomes. To test this possibility, we manually examined the region around the most frequently cleaved base pair on each chromosome (e.g., *Figure 2E,G,I,K*) but detected no double peak pattern similar to that seen over centromeres.

We next exhaustively tested the uniqueness of the centromere pattern. A profile was constructed from an ungapped alignment of centromeres of similar length (117–120 bp), and the average count at every position in the profile was used to scan every ungapped alignment in the genome (median

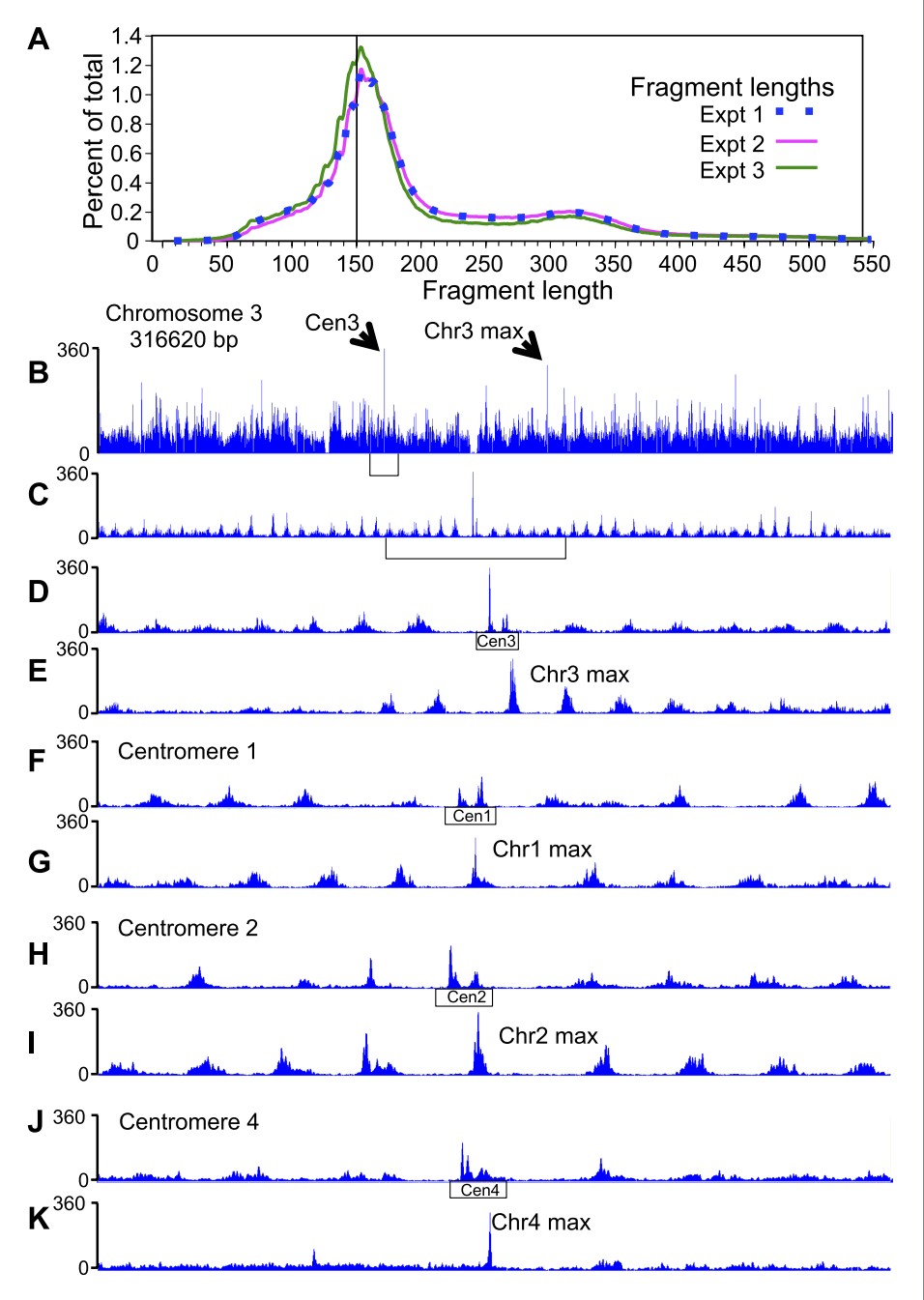

**Figure 2**. Pairs of cleavage clusters occupy CDEs. (**A**) Fragment length distributions for three experiments. For each length in base pairs, the percentage of the total number of mapped fragments is plotted. Experiment 1 (using Phusion DNA polymerase): 27,871,803 fragments; Experiment 2 (using KAPA polymerase): 62,926,977 fragments; Experiment 3: 81,473,515 fragments (biological replicate using KAPA). (**B–K**) Examples of cleavage profiles around centromeres, where tracks represent the total number of fragment ends within successive 10-bp windows. (**B**) Chromosome 3 profile, where the position of the centromere and of the most frequently cleaved nucleotide position on the chromosome are indicated. (**C**) Expansion of the region indicated by the bracket in (**B**). (**D**) Expansion of the region indicated by the bracket in (**C**), where the extent of Cen3 is indicated. (**E**) Expansion of the region around the most frequently cleaved nucleotide position at the same scale as in (**D**). (**F**) and (**G**) Same as (**D**) and (**E**), respectively, for Chromosome 1. (**H**) and (**I**) Same as (**D**) and (**E**), respectively, for Chromosome 2. (**J**) and (**K**) Same as (**D**) and (**E**), respectively, for Chromosome 4. (See also *Figure 2—figure supplement 1*).

*Figure 2. Continued on next page*

*Figure 2. Continued*
The following figure supplements are available for figure 2:
**Figure supplement 1**. The log-phase doubling rate of the H4S47C strain is normal.

764,931 alignments per chromosome). We scored alignments as the Pearson correlation (r) between the profile and the chromosome. For each chromosome, a centromere alignment to the profile (median r = 0.76) scored above the best non-centromere alignment (median r = 0.40) (*Table 1*). Removing each centromere from the model and rescanning ('delete-one jackknife') reduced their scores accordingly, but the median correlation (r = 0.52) still scored above the median correlation for the best non-centromeric alignment (r = 0.41). We conclude that the conspicuously different cleavage pattern at centromeres is unique to centromeres.

## Biased recovery of CDEII-containing fragments is attributable to high A+T content

Each double-strand cleavage should generate two fragments, and if they are recovered with equal probability, then the expected frequency of cleavage at a site for the left paired end should be the same as that for the right paired end. However, at all 16 centromeres, we observed a strong bias against recovery of the right-end fragment for the cleavage cluster on the left (on the CDEI side of CDEII) and against recovery of the left-end fragment for the cleavage cluster on the right (on the CDEIII side of CDEII) (*Figure 3A,C*). In contrast, there was no consistent recovery bias seen for the single peaks of the most frequently cleaved base pair on every chromosome when similarly aligned (*Figure 3B,D*). We suspected that the presence of ~40 bp more of the >90% AT-rich CDEII sequence on a fragment caused it to be strongly discriminated against during some step in the sequencing pipeline. Consistent with this possibility, we found that when fragments are aligned around the mid-Cen and mapped as normalized counts, in addition to the strong depletion directly over CDEII, there is a gradient of decreasing depletion with distance from the centromere (*Figure 3E*, red line). This gradual decrease with distance is expected if the fragments that are preferentially lost are those that span most of CDEII.

To ascertain whether the preferential loss of CDEII-spanning fragments is attributable to the well-known bias against the most AT-rich sequences in Solexa sequencing (*Bartfai et al., 2010*; *Lopez-Barragan et al., 2010*), we compared the recovery of CDEII-containing fragments to the recovery of non-centromeric fragments of similar AT-richness and length. There are 46 segments in the budding yeast genome that are at least 90% AT-rich over 83-bp, the median length of CDEII, and all of them are under-represented relative to expectation in our cleavage datasets (*Figure 3E*). Importantly, the degree of representation decreases significantly with increasing AT-richness between 90–100% (r = 0.52), showing depletions similar to what is seen for CDEII-containing fragments (*Figure 3F*). We therefore attribute the preferential loss of CDEII-containing fragments to their AT-richness, and not to any other feature of centromeric DNA. We observed similar losses for a Solexa library produced using Phusion DNA Polymerase, which has been reported to bias against such strongly AT-rich sequences (*Bartfai et al., 2010*; *Lopez-Barragan et al., 2010*), and libraries produced using KAPA DNA Polymerase, which has been engineered to be exceptionally processive on sequences that are strongly compositionally biased (*Quail et al., 2012*). It is more likely that an inherent feature of the Illumina platform, such as PCR-based cluster generation, is responsible for discrimination against sequencing templates that are >90% AT-rich (*Bartfai et al., 2010*).

## Centromere cleavage maps are incompatible with octasomes or tetrasomes

When centromere cleavage maps were displayed at base-pair resolution, we observed that each of the two cleavage clusters resolved into cleavage pairs separated by ~10 bp (*Figure 4A–F*). A composite alignment of all 16 centromeres centered over the midpoint of the functional centromere (mid-Cen) revealed that the inner peak maxima were 33-bp apart and the outer peak maxima were 53-bp apart (*Figure 4A*). No such distinct spacings were seen for the most frequently cleaved base pairs on each chromosome (*Figure 4G*). Because centromeres range in length from 111 bp for Cen4 to 120 bp for Cen12, we pooled aligned centromere cleavage data within length classes. For centromere lengths of

**Table 1.** Profile scanning for the centromere-specific cleavage pattern*

| Chr | # aligned | 15 centromere Profile | | | Delete-one jackknife† | | |
|---|---|---|---|---|---|---|---|
| | | # pass filters | First Cen | First FP | # pass filters | First Cen | First FP |
| 1 | 230,107 | 6 | 0.778‡ | 0.317 | 5 | 0.745 | 0.303 |
| 2 | 813,073 | 34 | 0.849 | 0.401 | 23 | – | 0.420 |
| 3 | 316,509 | 8 | 0.822‡ | 0.388 | 7 | 0.571 | 0.426 |
| 4§ | 1,531,822 | – | – | – | 31 | – | 0.394 |
| 5 | 576,763 | 51 | 0.807 | 0.611¶ | 48 | – | 0.625¶ |
| 6 | 270,050 | 14 | 0.726‡ | 0.389 | 13 | 0.723 | 0.306 |
| 7 | 1,090,829 | 33 | 0.756‡ | 0.450 | 32 | 0.707 | 0.466 |
| 8 | 562,532 | 7 | 0.459‡ | 0.346 | 4 | – | 0.293 |
| 9 | 439,777 | 11 | 0.493 | 0.375 | 2 | – | −0.092 |
| 10 | 745,640 | 22 | 0.626‡ | 0.402 | 19 | 0.579 | 0.357 |
| 11 | 666,705 | 23 | 0.634‡ | 0.407 | 15 | – | 0.326 |
| 12 | 1,078,066 | 48 | 0.742‡ | 0.430 | 47 | 0.698 | 0.453 |
| 13 | 924,320 | 48 | 0.873‡ | 0.491 | 53 | 0.847 | 0.498 |
| 14 | 784,222 | 23 | 0.778‡ | 0.398 | 16 | 0.748 | 0.384 |
| 15 | 1,091,180 | 21 | 0.796 | 0.457 | 11 | – | 0.420 |
| 16 | 947,955 | 32 | 0.764‡ | 0.455 | 23 | 0.464 | 0.478 |
| Median** | 764,931 | 22.5 | 0.760 | 0.402 | 17.5 | 0.518 | 0.407 |

*Alignments to the profile with more than three positions greater than three standard deviations from the mean of the profile position or with a maximum position less than the smallest maximum position within the profile (186) were excluded (filters). Pearson correlation coefficients are shown.
†Only jackknife results for the centromere deleted from the model are shown.
‡Multiple high-scoring centromere hits above the first false positive (FP) one or two base pairs apart.
§Cen4 (111 bp) was not included in the model.
¶Single base-pair cleavage peak at a site of anomalously low nucleosome occupancy.
**Medians are based on all alignments for all 16 chromosomes, whether or not they passed the filters.

117, 118, 119 and 120, we found that the distance between the inner peak maxima was 33 bp and between the outer peak maxima was 53 bp (*Figure 4C–F*). However, for the 111 bp Cen4, the inner peak maxima were ~25 bp apart and the outer peak maxima were ~45 bp apart (*Figure 4B*). Similar peak maxima were seen for gel-purified OP-labeled Cse4/H4S47C/H2A/H2B hemisomes reconstituted with a 78-bp Cen4 CDEII DNA duplex (*Furuyama et al., 2013*; *Codomo et al., 2014*) that had been subjected to in vitro cleavage reactions (*Figure 4—figure supplement 1*).

Similar 10-bp peak-to-peak distances within cleavage clusters were observed for all 16 centromeres, both for clusters on the left and on the right of the mid-Cen position. As explained below, our structural model cannot account for 10-bp distances between single H4S47C-anchored cleavages, which implies that these cleavages represent independent particles that are rotated by 10-bp relative to one another. Such 10-bp spacings between H4S47C-anchored cleavages have previously been interpreted as differences in rotational phasing (*Brogaard et al., 2012b*).

We can use these measurements to distinguish among the various structural models that have been proposed for the Cse4 nucleosome. The high-resolution crystal structure of the cenH3 octasome shows an arrangement of H4 residues, including S47, that is virtually identical to that for the conventional H3 octasome (*Tachiwana et al., 2011*), and so would predict a similar cleavage pattern over the centromere as is seen for highly occupied and phased nucleosomes genome-wide. This is clearly not the case at any yeast centromere, as even the minimal spacing of the inner cleavages is much larger than can be explained by our molecular dynamics model for a single nucleosome (*Figure 1*) or is observed for highly occupied and phased H3 nucleosomes throughout the genome (*Figure 4H*). Likewise, Cse4/H3 octasomes and (Cse4/H4)$_2$ tetrasomes also predict the same mirror-image symmetrical arrangement of two H4s as canonical octasomes, and so our data also exclude those models.

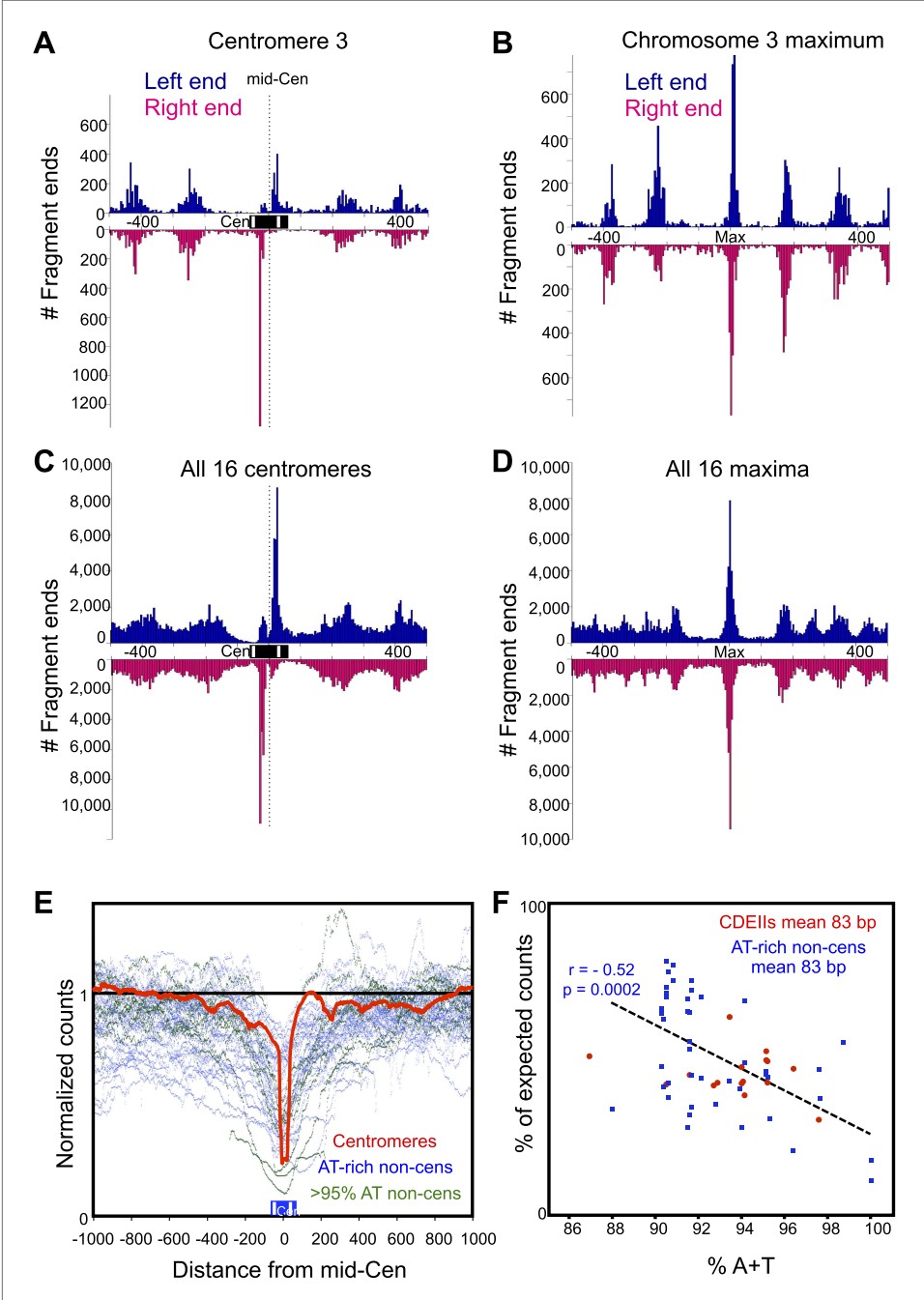

**Figure 3**. Fragments spanning the mid-Cen are depleted due to high A + T content. (**A**–**D**) Histograms of cleavage counts in successive 10-bp intervals are plotted separately for left (blue above) and right (magenta below) ends of mapped fragments. (**A**) The 1-kb interval around the mid-Cen of Cen3. Dotted line marks the mid-Cen position. (**B**) The 1-kb interval around the most frequently cleaved nucleotide position on Chromosome 3. (**C**) Same as (**A**) for all 16 aligned mid-Cens. (**D**) Same as (**B**) for all 16 aligned most frequently cleaved nucleosome positions. (**E**) Fragments were stacked over aligned mid-Cens, and normalized counts for each base-pair position were calculated (red line), where a value of 1 represents the random expectation. For comparison, normalized count plots are superimposed for the 40, 83-bp ≥90.5% AT-rich intervals (dots), where data for intervals 90–95% AT-rich are plotted in blue and intervals >95% AT-rich are plotted in green. Strong depletion is seen directly over the middle of aligned CDEIIs, gradually approaching 1 with distance from the mid-Cen. Similarly, strong depletion with gradual approach to 1 is also seen for all 83-bp AT-rich control sequences. (**F**) The percentage of expected number of cleavage sites within the 16 CDEII sequences, comprising a median of 83 bp, and that of the 46 sequences ≥90% A+T are plotted

*Figure 3. Continued on next page*

*Figure 3. Continued*

as a function of AT-richness. Depletion relative to expectation is seen for CDEIIs, indicating that the bias against recovery can be accounted for by high AT-richness.

---

This leaves two other models, a version of the (Cse4/H4/Scm3)$_2$ particle and the hemisome, as potentially explaining the cleavage patterns that we observe. The presence of Scm3 as an integral part of the Cse4 nucleosome has been excluded based on subsequent in vivo mapping (*Shivaraju et al., 2011*) and in vitro reconstitution studies (*Cho and Harrison, 2011*; *Xiao et al., 2011*), which leaves the hemisome as the only model that can account for the cleavage patterns that we observe. Specifically, a model in which independent hemisomes align around the mid-Cen position in either orientation and in two rotational phases can account for all of the cleavage peaks that we observe over centromeres.

Although 15 centromeres show nearly precise alignment to the functional centromere, with ~10-bp/33-bp/10-bp spacings between peak maxima, Cen4 is an exception in showing ~10-bp/25-bp/10-bp spacings (*Figure 4—figure supplement 2*). This ~8-bp difference in cleavage pattern corresponds to the fact that, as pointed out above, Cen4 is 6–9 bp shorter than the other 15 centromeres. A parsimonious explanation of this correspondence is that the position of H4 residue 47 in the Cse4 core particle is fixed relative to the nearer junction regardless of whether the junction is CDEI-II or CDEII-III. Thus, cleavages on the left side of the mid-Cen position would preferentially occur a fixed distance from CDEI-II, whereas cleavages on the right side of the mid-Cen position would preferentially occur a fixed distance from the CDEII-III junction. In both cases, the 10-bp spacing between cleavage maxima implies that the rotational phase of the Cse4 nucleosome is established by where the nearer sequence-specific DNA-binding protein binds, whether it is Cbf1 at CDEI or the CBF3 complex at CDEIII.

## Single H4-containing particles are responsible for centromere cleavage distances

We next compared experimental cleavage distances to our simulated cleavage model, taking into account the Solexa library end-polishing procedure, which removes 3′ overhangs and fills in the complement to 5′ overhangs (*Figure 1C*). At least one cleavage on the Watson strand and one on the Crick strand are required to observe a fragment by sequencing. Cu$^+$ attack on DNA results in base-loss, and we would observe the adjacent position when the cleaved products are sequenced after Solexa end-polishing. We took an unbiased approach and asked what are the preferred distances between left and right ends of fragments observed experimentally. The left end of a fragment marks the position of a cleavage and base loss on the Watson strand and the right end of a fragment marks the position of a cleavage and base loss on the Crick strand. The distances between left and right ends thus reflect preferred cleavage locations. We observed peaks at −2, +5 and +12 in the Watson-Crick (W-C) distributions (*Figure 5A*, red curve), which are all explained by the DNA positions shown to be accessible to H4S47C-phenanthroline-Cu$^+$ in our structural model (*Figure 5—figure supplement 1*). Additionally, the structural model helps us distinguish between W-C peaks due to cleavages by H4S47C on one side of the dyad and W-C peaks due to cleavages on two H4S47Cs across the dyad. The +5 W-C peak results from cleavages by the same H4, whereas the −2 and +12 W-C peaks result from cleavages occurring on both sides of the dyad.

An alternative way to distinguish between cleavages by a single H4 and cleavages by two H4s across a dyad is to measure distances between the left ends of fragments, W–W′, and the right ends, C–C′. Their genome-wide distributions revealed strong peaks at +1 and +7 (*Figure 5B,C* red curves). The peak at +1 reflects cleavages due to the same H4 molecule on the same side of the dyad, while the peak at +7 reflects cleavages due to adjacent H4 molecules across the nucleosome dyad. Thus, the predicted cleavage sites imply that the fragment spacings observed genome-wide resulted from a combination of cleavages by H4 on both sides of the nucleosome dyad axis.

In order to determine whether our simulated cleavage model can explain the patterns at centromeres, we generated W-C distributions for reads that are within 125 base-pairs of the center of CDEII (*Figure 5A*, black curve). Whereas we observed a peak at +5, which can arise from the presence of a single H4, the peaks at −2 and +12, which require two H4s on either side of the dyad axis, were not seen. It might be argued that we do not observe −2 and +12 peaks in the W-C distributions near CDEII

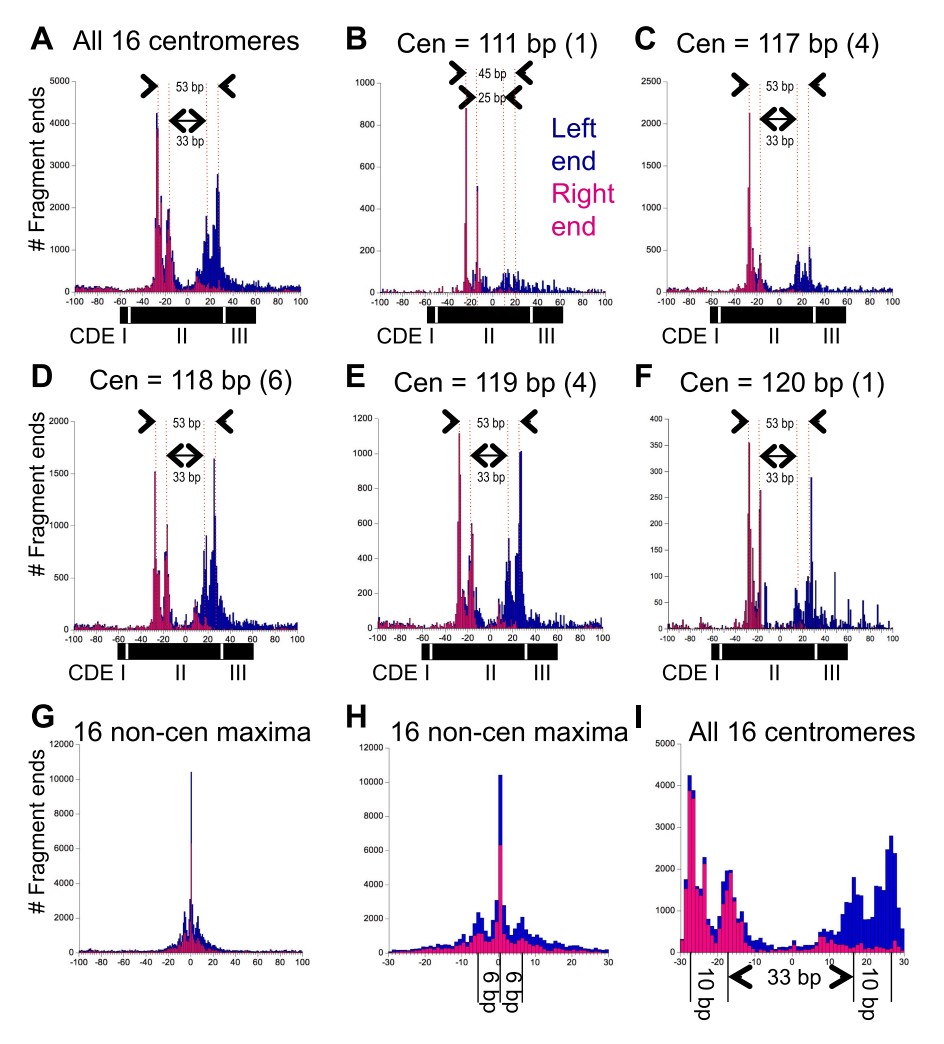

**Figure 4**. Pairs of 10-bp cleavage sites are symmetrically offset around the mid-Cen. (**A**) Composite histogram of cleavage counts for all fragments in successive 1-bp intervals for all 16 centromeres. Blue (left end) and magenta (right end) bars are stacked, such that the overall profile represents the sum at each base-pair position. (**B**–**F**) Same as (**A**) grouped by centromere size class as indicated. (**G**) Same as (**A**) for the most frequently cleaved nucleotide position on each of the 16 chromosomes. (**H**) Expansion of (**G**) showing the 6-bp average spacing between observed cleavages. (**I**) Expansion of (**A**) showing the 10-bp/33-bp/10-bp average spacing between cleavages observed over centromeres. See also *Figure 4—figure supplement 1* and *Figure 4—figure supplement 2*.
The following figure supplements are available for figure 4:

**Figure supplement 1**. Cleavage mapping of Cse4 hemisomes reconstituted on Cen4 CDEII DNA in vitro.

**Figure supplement 2**. Cen4 spacing around the mid-Cen position is anomalously short.

because of preferential loss of AT-rich fragments spanning CDEII (*Figure 3E,F*). However, in that case, we would also not see the +5 peak, suggesting that the loss of −2 and +12 peaks is not due to an AT-rich sequencing bias. To verify this interpretation, we examined same-strand (W–W' and C–C') distributions, which obviates the need to recover fragments spanning CDEII. For the W–W' and C–C' distributions we again observed a strong peak at +1 (*Figure 5B,C*, black curves), which can arise from a single H4, whereas the peak at +7, which results from cleavage across a dyad axis, was absent. Rather, the 10-bp peak spacing evident in the centromere maps (*Figure 4*) and attributable to rotational phasing was a prominent feature in both W–W' and C–C' for centromeres, but not genome-wide.

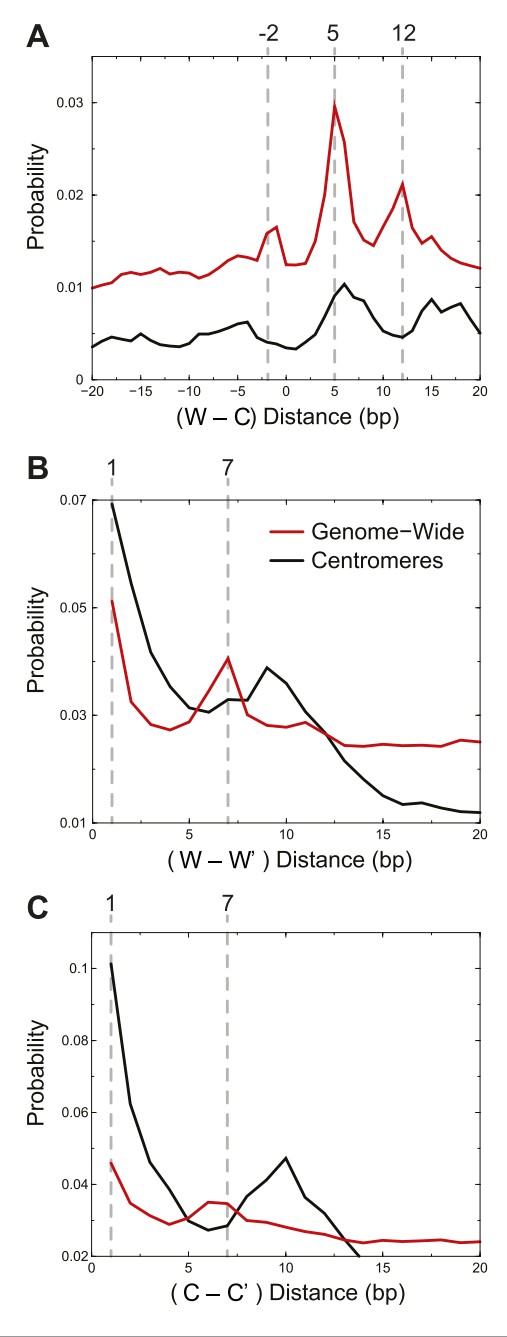

**Figure 5**. Centromere cleavage distances are predicted by a single-H4 model. Distances between fragment ends are plotted for all fragments (Genome-wide) and for fragments mapping to within 125 bp of the center of CDEII of all chromosomes (Centromeres). Distribution of the distances between (**A**) right and left fragment ends (W–C); (**B**) left fragment ends (W–W'); (**C**) right fragment ends (C–C'). Gray dashed lines mark the peaks observed in the genome-wide distributions. See also *Figure 5—figure supplement 1*.
*Figure 5. Continued on next page*

Taken together, the absence of peaks representing H4S47C-anchored cleavages across the dyad axis but the presence of peaks representing a single H4 at centromeres provides independent confirmation based on mapping of cleavage positions that single H4-containing particles predominate at centromeres.

## Oppositely oriented hemisomes are present on different sequences in the population

The simplest explanation for the cleavage patterns around the mid-Cen is that oppositely oriented hemisomes occupy different members of the cell population. However, the cleavage data alone do not exclude a model in which two oppositely oriented hemisomes flank the mid-Cen. To examine this possibility, we superimposed the H4S47C cleavage data with MNase-seq and Cse4 ChIP/input profiles, using a variation of the 'V-plot' method that we introduced to facilitate the precise determination of particle position and size for MNase-seq and other paired-end sequencing data (*Henikoff et al., 2011*). In our original implementation, a V-plot consists of a dotplot representation in which a dot is placed on the X-axis for the position of a fragment midpoint relative to a fixed position and on the Y-axis at a position representing the length of the fragment. For H4S47C-anchored cleavage mapping, we are interested in plotting the precise left and right end positions, and so we generate two V-plots, one in which the X-axis position corresponds to the left end of a fragment and one in which it corresponds to the right end. When plotted in this way, with the mid-Cen as the fixed position on the X-axis and increasing fragment length on the Y-axis, we observe vertical lines of dots in pairs and corresponding pairs of diagonal lines that represent preferred cleavage sites at the same cleavage position on fragments of increasing length (*Figure 6*, *Figure 6—figure supplement 1*).

Several notable features are revealed by the left-right V-plots. First, the two double vertical lines of dots seen on either side of the mid-Cen mark the location of two 10-bp cleavage pairs observed using conventional cleavage density histograms. Second, the fact that the vertical lines are densely populated up to ~400 bp excludes the possibility that the deficiency of the AT-rich mid-CDEII-containing fragments is attributable in part to being on average 40-bp longer than their sister fragments and so discriminated against based on size. Third, clusters of left and right fragments on either side of the centromere,

*Figure 5. Continued*

The following figure supplements are available for figure 5:

**Figure supplement 1**. Examples of predicted cleavage distances.

representing neighboring nucleosomes, align closely with MNase-seq nucleosomal profiles for all 16 centromeres, and show centromere-to-centromere variability as previously observed (*Cole et al., 2011*; *Krassovsky et al., 2012*). Fourth, each left end must pair with a right end, and the fact that for each left-end vertical there is a corresponding dense right-end diagonal and vice-versa implies that there are dense cleavages on either side of the centromere in addition to those expected from cleavages centered over neighboring nucleosomes, a point that we will return to in the next section. Finally, when log$_2$(Cse4/Input) ChIP-seq profiles are superimposed over the V-plots, we see that the midpoints of the vertical lines of dots marking the preferred cleavages are precisely centered over the position of maximum occupancy of the Cse4 nucleosome.

To confirm that cleavage features we observed depended on the H4S47C mutation, we subjected wild-type cells to OP-labeling and cleavage reactions. The cleavage frequency in a wild-type strain is too low to obtain a library comparable to that of the H4S47C strain, which requires that two cleavages are sufficiently close to generate <500-bp fragments for paired-end sequencing. Therefore, purified DNA was cleaved to completion with AluI, which cleaves the sequence AG^CT and leaves blunt ends. The very low level of cleavages observed over centromeres (*Figure 6—figure supplement 2A*) confirms that centromere cleavage patterns are specific for H4S47C and not for centromere-specific DNA-binding proteins. As an additional control, we identified background cleavages by isolating DNA from untreated cells and digesting with AluI before library preparation. This revealed that the level of background cleavage over centromeres is similarly low regardless of whether or not cells were subjected to OP-labeling and cleavage reactions, as expected if centromeres are fully occupied and protected during the cleavage reaction (*Figure 6—figure supplement 2B*).

## Cse4 hemisomes are stable at centromeres

CenH3 nucleosomes are hypersensitive to MNase digestion in vivo (*Takahashi et al., 1992*; *Dalal et al., 2007*; *Krassovsky et al., 2012*), and some authors have suggested that hemisomes might represent unstable intermediates in the assembly or disassembly of octasomes (*Black and Cleveland, 2011*; *Dunleavy et al., 2013*). H4S47C-anchored cleavage mapping relies on nucleosomes remaining stably wrapped during the multistep nuclear labeling and washing procedures, which includes an overnight incubation with OP-labeling reagent that is performed prior to the actual Cu-dependent cleavage reaction. Therefore, our mapping of Cse4 hemisomes over centromeres at levels expected for such high AT-rich sequences (*Figure 3E*) implies that they are stable particles in vivo as we previously showed for hemisomes wrapped by CDEII in vitro (*Furuyama et al., 2013*). Nevertheless, it remained formally possible that the hemisomes that we mapped at centromeres were generated by dissociation of Cse4 octasomes during the OP-labeling step.

To test whether OP-labeling conditions promote the dissociation of octasomes, we reconstituted octasomes with either H3-H4 or Cse4-H4S47C and with either the 147-bp 601 positioning sequence or with a 147-bp Cen3-containing DNA segment. Using a gel-shift assay, we observed no changes in particle migration associated with the OP-labeling procedure for any of the reconstituted octasomes (*Figure 7A*). To confirm that the particles undergoing a gel-shift remained octasomes during this procedure, we gel-purified reconstituted particles with or without the OP-labeling treatment and performed atomic force microscopy (AFM). We found that OP-treated and untreated Cse4 and H3 octasomes are similar in height to one another but are ~40–50% taller than Cse4-H4S47C/CDEII hemisomes reconstituted and incubated in parallel (*Figure 7B*), consistent with previous studies (*Furuyama et al., 2013*; *Codomo et al., 2014*; *Walkiewicz et al., 2014*). We conclude that the OP-labeling procedure does not alter the form of nucleosomes reconstituted with centromeric DNA.

To confirm that OP labeling does not alter centromeric nucleosomes in vivo, we profiled wildtype and H4S47C cells genome-wide before and after the OP-labeling treatment. Using modified MNase-seq (*Henikoff et al., 2011*), we determined that the position of the Cse4 nucleosome is the same in H4S47C and wild-type and in OP-treated and untreated cells (*Figure 7C*). To confirm that centromeric chromatin is intact after OP-labeling, we also profiled MNase protection of insoluble chromatin (*Figure 7—figure supplement 1*), which is ~100-fold enriched for centromeric chromatin (*Krassovsky et al., 2012*), and performed ChIP-seq on the soluble fraction using a Cse4-specific

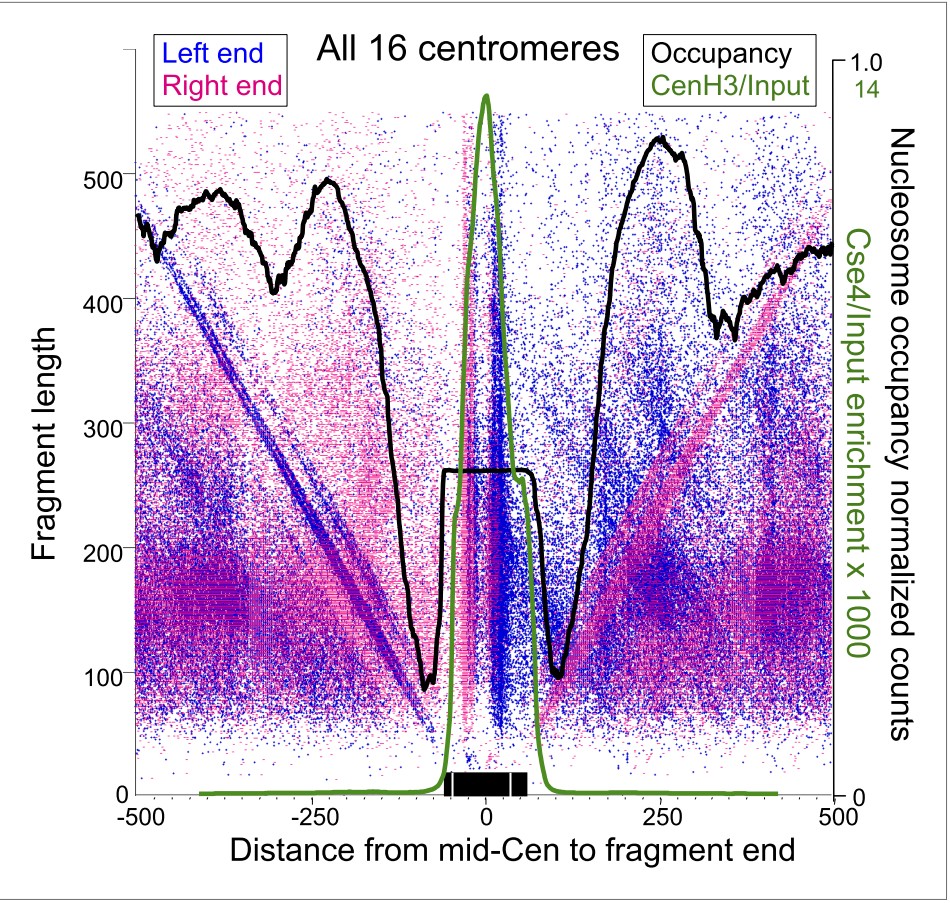

**Figure 6**. Paired cleavage sites on opposite sides of the Cse4 nucleosome over CDEII. Left-right V-plots with MNase and Cse4 ChIP/Input profiles superimposed for all centromeres aligned over the mid-Cen. The X-axis position for each blue dot corresponds to the left end of a mapped fragment and the X-axis position for each red dot corresponds to the right end. With the mid-Cen as the fixed position on the X-axis and increasing fragment length on the Y-axis, we observe vertical lines of dots in pairs representing preferred cleavages within CDEII. The blue and red diagonals respectively correspond to (mostly background) cleavages on the left and right sides of the centromere. See also *Figure 6—figure supplements 1 and 2*.

The following figure supplements are available for figure 6:

**Figure supplement 1**. Left-right V-plots with MNase and Cse4 ChIP/Input profiles superimposed for all 16 centromeres.

**Figure supplement 2**. Left-right V-plot representation of likely in vivo cleavages over all 16 centromeres in a wild-type strain.

antibody (*Figure 7—figure supplement 2*). In both cases, midpoint V-plots reveal that CDEI, CDEII and CDEIII are strongly MNase-protected relative to flanking regions in both strains with or without OP treatment in both soluble and insoluble chromatin fractions. This extent of Cse4 particle occupancy is half that necessary to accommodate two spaced hemisomes, because if the cleavage pairs were attributable to two hemisomes particles wrapping ~60 bp with spacings of 33–53 bp between cleavage sites, then the total span would be 153–173 bp, or about twice the ~80 bp that the Cse4-containing particle occupies. This leaves the single hemisome per centromere model as the only one that can account for our cleavage data.

To further test this conclusion, we investigated the basis for cleavages around the centromere that do not correspond to neighboring nucleosomes. That is, the fact that there are fairly uniform diagonals extending from ~50 bp up to >400 bp is not consistent with cleavages occurring only at the center of

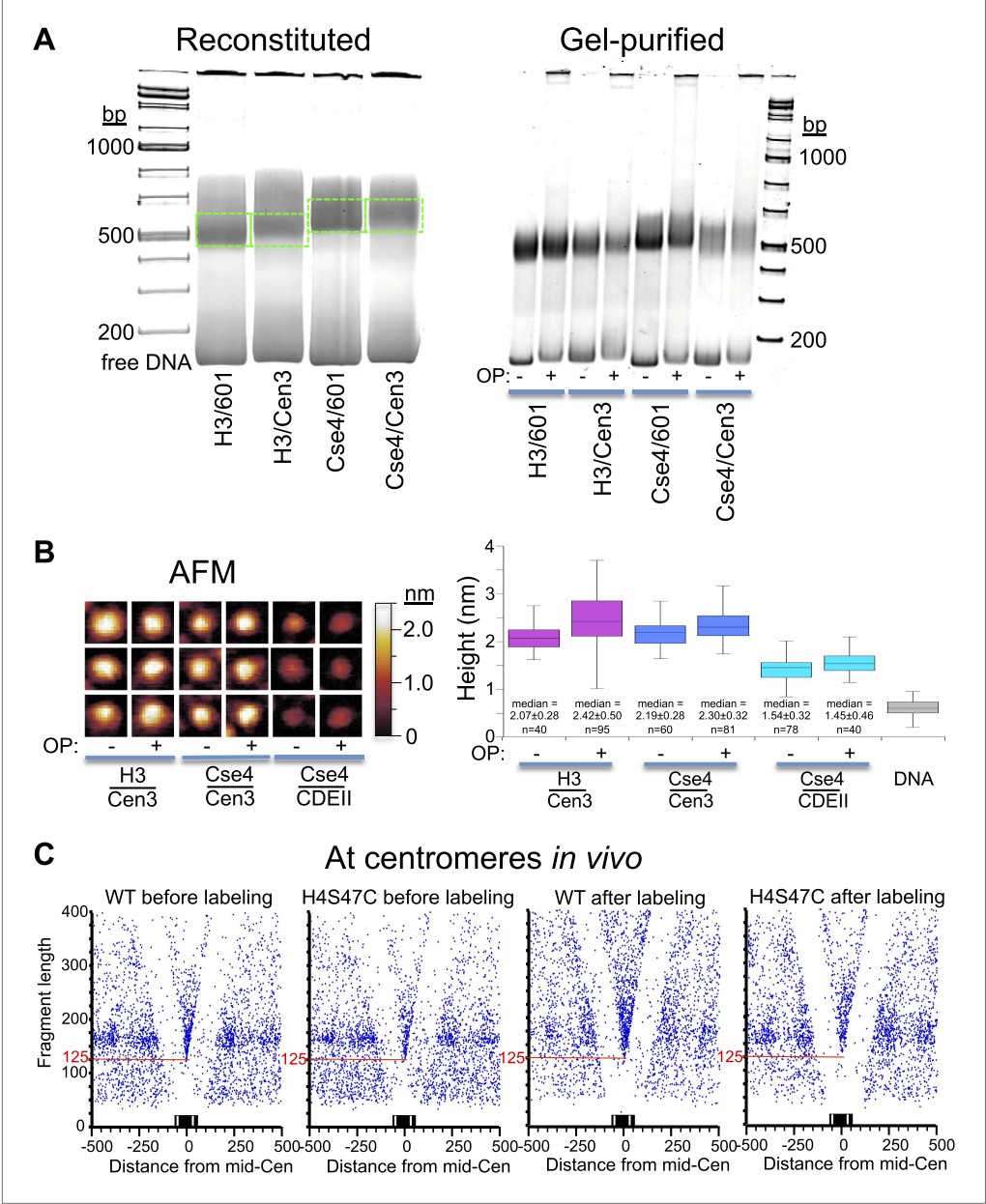

**Figure 7**. OP-labeling does not disrupt reconstituted or native nucleosomes. (**A**) Octasomes were reconstituted by salt dialysis on 147-bp DNAs, subjected to native PAGE (left), and gel-shifted bands were excised (dotted green lines) and extracted as described (*Codomo et al., 2014*). OP-treated and untreated gel-purified particles were subjected to native PAGE (right). No instability of Cse4/Cen3 octamers (*Dechassa et al., 2011*) was observed during low-temperature incubation during storage at 4°C (*Xiao et al., 2011*; *Furuyama et al., 2013*). Using the intensity ratio of the gel-shifted band to the free DNA band as a measure of octasome stability, stability in the presence or absence of OP reagent was similar in each case (Fraction +OP/-OP: H3/601: 0.8; H3/Cen3: 1.0; Cse4/Cen3: 1.1; Cse4/Cen3; 1.0), based on the average of two determinations. (**B**) AFM analysis of reconstituted particles ± OP treatment. Three representative particles from each sample are shown on the left at the same magnification and dynamic range, where the height was set at 0.6 nm below the mean height of free DNA imaged in the same scan. The median heights for 1 min trypsinized OP-treated and untreated H3 and Cse4 octasomes and Cse4 hemisomes are similar to what we previously reported using the same protocol (2.10 nm for Cse4/Cen3 octasomes and 1.59 nm for Cse4/CDEII hemisomes [*Codomo et al., 2014*]). (**C**) MNase-seq was performed on wildtype (WT) and H4S47C mutant cells ± OP labeling. The mid-CDE position of each of the 16 centromeres was aligned at zero on the X-axis position. A blue dot corresponds to the midpoint of each mapped fragment on the X-axis and the fragment length on the Y-axis.
*Figure 7. Continued on next page*

*Figure 7. Continued*

The sharp vertex located over the mid-Cen marks the X-axis position of the minimally protected fragment and its Y-axis position (red line) indicates its length. See also *Figure 7—figure supplement 1-5*.

The following figure supplements are available for figure 7:

**Figure supplement 1**. MIdpoint V-plot representations of MNase-seq generated fragments from insoluble chromatin over all 16 centromeres.

**Figure supplement 2**. MIdpoint V-plot representations of Cse4 ChIP fragments over all 16 centromeres.

**Figure supplement 3**. The fragile nucleosome over the Gal1-10 UASg is not detected by H4S47C-anchored cleavage mapping.

**Figure supplement 4**. Fragile nucleosome are not detected by H4S47C-anchored cleavage mapping.

**Figure supplement 5**. Nucleosome-depleted regions show preferential background cleavages.

flanking nucleosomes, but rather suggests that these are caused by mostly background cleavages on either side of the centromere. As the hydroxy radical cleavage protocol that we use was originally developed for mapping non-specific cleavages within linker DNA (*Cartwright and Elgin, 1982*), these diagonals indicate a moderate level of background cleavage between flanking highly occupied and phased nucleosomes.

We also observed background cleavages at sites of 'fragile' nucleosomes, which are thought to be unstable based on hypersensitivity to MNase (*Xi et al., 2011*). The best-studied fragile nucleosome is over the Gal4 UAS in the Gal1-10 regulatory region, when yeast are grown in glucose (*Floer et al., 2010*) (*Figure 7—figure supplement 3A*). We therefore expected to observe high frequency cleavage by this nucleosome in our cleavage datasets. However, there was no evidence of nucleosome-directed cleavage, but rather especially strong cleavage diagonals directly over the Gal4 UAS (UASg, *Figure 7—figure supplement 3B*). We observed similarly strong Xs over other fragile nucleosomes (*Figure 7—figure supplement 4*). The 'X' pattern of depletion seen at these sites resembles that for known nucleosome depleted regions, such as those around binding sites for the Reb1 and Abf1 transcription factors (*Figure 7—figure supplement 5*). We further confirmed this interpretation by mapping AluI-generated fragments of DNA in the Gal1-Gal10 region from wildtype cells subjected to OP-labeling and cleavage reactions. We observed a strong diagonal representing the left end of fragments cleaved by AluI at a site on the right side of the Gal4 UAS, but only weakly extending beyond the edge of the well-occupied next nucleosome to the left (*Figure 7—figure supplement 3C,D*). In contrast, centromeres showed very little cleavage relative to regions on either side (*Figure 6—figure supplement 2A*), indicating protection from background cleavages and confirming that Cse4 nucleosome-directed cleavages depend on the H4S47C mutation. The stability of the Cse4 centromeric nucleosome and protection from background cleavages under these conditions confirm that Cse4 hemisomes are immobile at centromeres, as expected from the stability of Cse4 hemisomes on short DNA fragments, including a fragment consisting of CDEII alone (*Figure 7B*; *Furuyama et al., 2013*).

## Discussion

We have used directed chemical cleavage mapping to determine the precise position of the single Cse4 nucleosome at each budding yeast centromere. Using molecular dynamics simulation of H4S47C-anchored cleavages, and verification of the resulting model based on cleavage patterns at highly occupied and phased nucleosomes, we find that the pattern over all 16 centromeres, and the distance between cleavage sites, is profoundly different from what is seen elsewhere in the genome. As octasome and tetrasome structures for the Cse4 nucleosome position two H4S47 residues immediately on either side of the dyad axis, just as for conventional H3 octasomes and tetrasomes, our cleavage data rule out these structures as making a substantial contribution to Cse4 occupancy at centromeres (*Figure 8*). Indeed, the only proposed structure that fits our data is a hemisome in one of four mutually exclusive orientations occupying each centromere within the population of cells. Thus, the Cse4 nucleosome shows remarkable rotational flexibility within the confines of the 78–86 bp CDEII sequence,

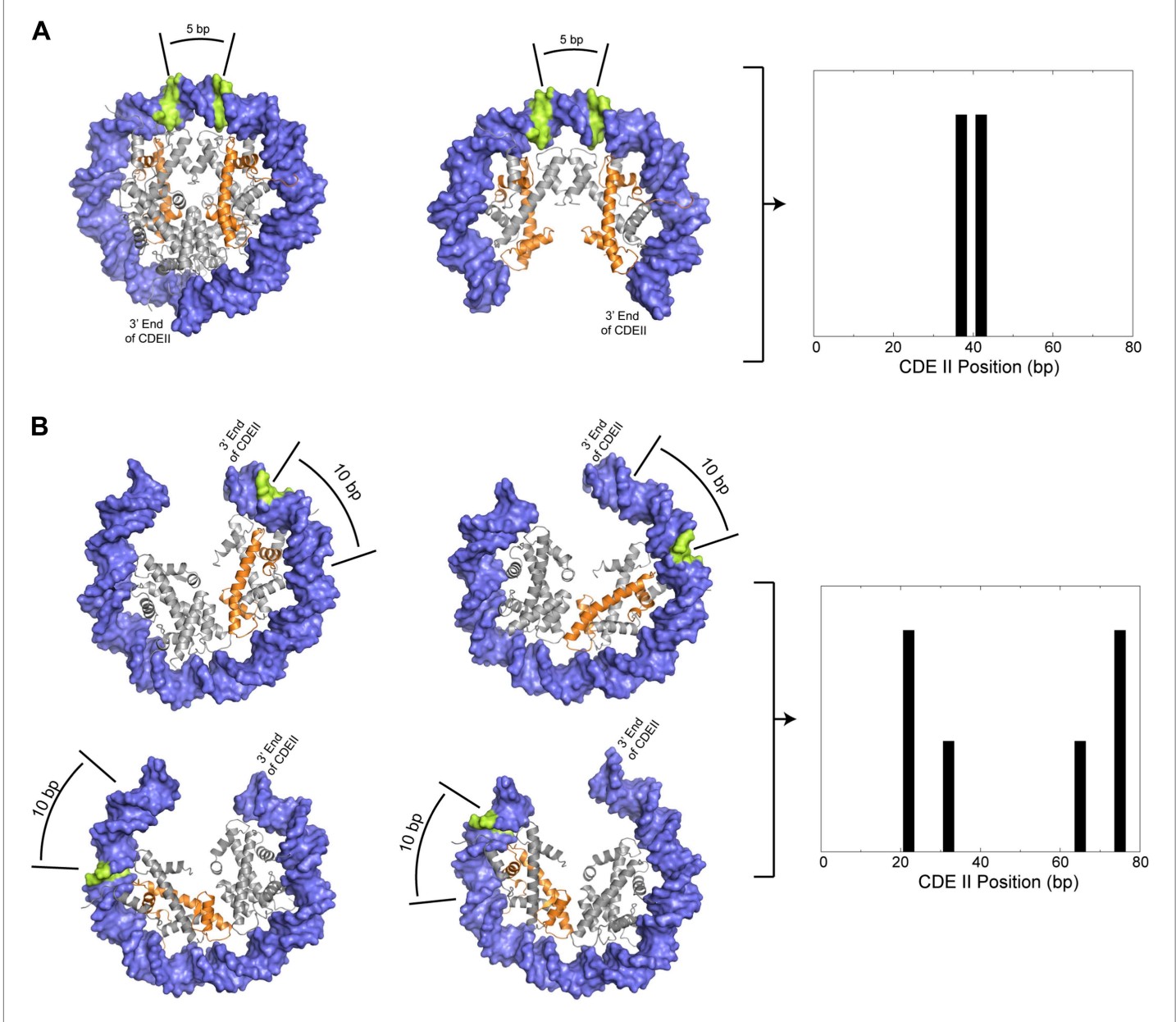

**Figure 8**. Preferred cleavage positions for proposed structural models of the CDEII nucleosome. (**A**) H4S47C-anchored cleavage reactions within an octasome (left) or an (H3/H4)$_2$ tetrasome (middle) centered over CDEII would give rise to mapped sites that are ~5 bp apart (right). (**B**) Observed cleavage positions (right) are explained by a structural model of a hemisome occupying either of two rotational positions (compare left vs middle models) in either of two reflectional orientations (compare upper vs lower models). The ~7 bp closer spacing of Cen4 cleavage pairs (*Figure 4—figure supplement 2*) implies that positioning is determined by the distance of H4S47C from the closer end of CDEII. Histone H4 is indicated in orange and preferred cleavage sites are indicated in green.

where it can assume opposite orientations and two preferred rotational phasing patterns. Such flexibility might be important for brief hemisome–octasome transitions reported for both budding yeast and human cenH3s (*Bui et al., 2012*; *Shivaraju et al., 2012*).

The flexibility in orientation and phasing that we observed for such a precisely positioned particle is especially remarkable considering that yeast centromeres are directional at the DNA level, with the >90% AT-rich CDEII sequence oriented between Cbf1-bound CDEI and CBF3-bound CDEIII. We attribute this rotational and reflectional flexibility to the fact that a 78-bp CDEII duplex is sufficient to stably wrap a hemisome in vitro (*Furuyama et al., 2013*), and that AT-richness alone is sufficient for full in vivo

function of CDEII, as some random 92% AT-rich sequences substituted for natural CDEII supported faithful segregation of mini-chromosomes (**Baker and Rogers, 2005**). This interpretation helps to clarify the roles of Cbf1 and the CBF3 complex in maintaining centromeres. Both proteins tightly bend DNA and thus delimit an ~80-bp sequence for assembly of the Cse4 nucleosome (**Henikoff and Furuyama, 2012**). CBF3 is essential for recruiting Cse4, where the Ndc10 subunit interacts with the Scm3 Cse4-specific chaperone (**Shivaraju et al., 2011**). In addition, our findings suggest that Cbf1 acts as a barrier on the left that sets the sharp rotational positions of the Cse4 nucleosomes that are asymmetrically oriented with H4S47 closest to CDEI. This implied role of Cbf1 in Cse4 hemisome positioning is consistent with our previous analysis of MNase-seq data in a *cbf1Δ* mutant strain showing that loss of Cbf1 is accompanied by a shift in position of the Cse4 nucleosome (**Krassovsky et al., 2012**). The rotational and reflectional flexibility of the Cse4 nucleosome implied by our study argues against specific roles for either Cbf1 or CBF3 in orienting the histone core within the single DNA wrap (**Xiao et al., 2011**).

Although budding yeast are especially suitable for application of directed chemical cleavage mapping, because replacing endogenous histones with the mutant H4S47C version is straightforward, one might envision applying this strategy to complex genomes. Complete replacement of the replication-coupled histone gene cluster has been accomplished in *Drosophila* (**Gunesdogan et al., 2010**), and conceivably other strategies for introducing cysteine-substituted histones might be considered for other model organisms. The application of this technology to the study of cenH3 nucleosomes more generally has the potential of resolving current debates as to the generality of hemisomes vs octasomes at centromeres. Current evidence suggests that both types of particles are present in animal genomes (**Bui et al., 2012**), although which are at functional centromeres and which are on chromosome arms has been unclear. MNase-seq and ChIP-seq have been applied to the problem, but particle size has been ambiguous because of the inherent sensitivity of hemisomes to MNase digestion (**Dalal et al., 2007**; **Krassovsky et al., 2012**) and ambiguities caused by partial unwrapping of octasomes (**Tachiwana et al., 2011**) and internal cleavages and nibbling (**Hasson et al., 2013**). Together with high-resolution native ChIP, directed cleavage mapping has the potential of unambiguously determining particle composition and structure in complex genomes.

## Materials and methods

### Structural model

We constructed a molecular model of cysteine-orthophenanthroline (OP) using Avogadro (**Hanwell et al., 2012**). We replaced the serine at position 47 of H4 in the yeast nucleosome structure (PDB ID:1ID3) with cysteine-OP. Our simulation system consisted of H3-H4(47C-OP) bound to 10 bp of DNA in the same conformation as in the nucleosome structure (corresponding to −2 to −12 positions with respect to the dyad). The DNA, H3 and H4 were held static, except for amino acid residues that were within 10 Å of H4S47C-OP. The simulation system was minimized and ~30,000 conformations of H4S47C-OP were generated using Chiron (**Ramachandran et al., 2011**).

### Yeast culture

Yeast strain Sby3 (*MATa bar1-1 ura3-1 leu2, 3-112 his3-11 trp1-1 can1-100 ade2-1*) was used as control cells for cleavage mapping, MNase-seq and ChIP as in our previous centromeric chromatin analysis (**Krassovsky et al., 2012**). Construction of H4S47C from BY4741 (*MATa his3Δ1 leu2Δ0 met15Δ0 ura3Δ0*) was previously described (**Brogaard et al., 2012b**). The two histone H4 genes (HHF1 and HHF2) had been deleted and replaced by a single HHF1 gene with a cysteine codon replacing the serine codon at position 47. Growth and viability measurements were performed using a Vi-Cell automated cell counter (Beckman–Coulter, Brea, CA). Plasmid loss assays were performed as described by **Koshland et al. (1987)**.

As previously reported (**Brogaard et al., 2012a**), H4S47C cells show growth anomalies relative to BY4741, which we have determined are caused by failure to achieve a quiescent state upon reaching stationary phase (*Figure 2—figure supplement 1*). This failure is likely attributable to deletion of HHF2, which results in reduced starvation resistance (**Davey et al., 2012**) and is associated with reduced lifespan (**Feser et al., 2010**).

### In vitro cleavage mapping

To map H4S47C-anchored cleavages within a reconstituted H3 octasome, octamers were prepared according to **Luger et al. (1999)** and OP-labeled by addition of a 10-fold molar excess of tris(2-carboxyethyl)phosphine (TCEP) for 10 min, then a 30-fold molar excess of OP reagent in DMSO in

the dark. Samples were incubated at room temperature 2 hr and at 4°C overnight, quenched with 1/350 1.4M β-mercaptoethanol, and excess OP-reagent removed with a Bio-Spin P-30 column (Bio-rad, Hercules, CA). Reconstitutions were performed with 601 DNA according to *Luger et al. (1999)*. Cleavage reactions proceeded by addition of 1 vol 5 mM NaCl, 100 mM Tris pH 7.5, 300 µM CuCl$_2$, mercaptopropionic acid to 6 mM, and H$_2$O$_2$ to 6 mM. After 20 min at room temperature, reactions were quenched with neocuproine in DMSO to 2.5 mM, and DNA was extracted using standard methods. SOLID sequencing was performed on the resulting fragments as previously described (*Brogaard et al., 2012b*).

To map H4S47C-anchored cleavages within Cse4 hemisomes, a 78-bp oligonucleotide with the sequence shown in *Figure 4—figure supplement 1*, that was 5′ end-labeled with 5′-Cy3 and 3′ end-labeled with 3′-AmC7-Q+Alexa488, was annealed with its unlabeled complementary unlabeled oligonucleotide. This 5′-Cy3-CDEII-Alexa488-3′ duplex was mixed with a fourfold excess of unlabeled CDEII duplex and used for octasome reconstitution and hemisome splitting To prepare OP-labeled hemisomes, Cse4/H4S47C/H2A/H2B octamers were lightly trypsinized as described (*Furuyama et al., 2013*) then subjected to in vitro OP-labeling and P-30 clean-up as described above, followed by reconstitution of pseudo-octasomes with CDEII DNA in 2M NaCl (*Furuyama et al., 2013*). After dialysis vs 4M urea to split pseudo-octasomes into two hemisomes, samples were electrophoresed on a 7% Tris-acetate (no EDTA) PAGE gel and bands were excised and eluted as described (*Codomo et al., 2014*), then subjected to in vitro cleavage reactions as described above. Reaction products from (1) the shifted band, which showed a strong gelFRET signal indicative of tight wrapping around a hemisome, (2) the unshifted (control) band and (3) G+A and C+T Maxam-Gilbert ladders produced from the 5′-Cy3-CDEII-Alexa488-3′ labeled oligonucleotide duplex, were electrophoresed on a 15% sequencing gel (*Sambrook et al., 1989*). The gel was scanned for Alexa488 and Cy3 using a Typhoon Trio and images were processed using ImageJ.

To ascertain the stability of reconstituted Cse4 octasomes and hemisomes, octamers were prepared as described (*Furuyama et al., 2013*) and reconstituted with 601, Cen3 or CDEII DNA by salt dialysis at 37°C as described by *Xiao et al. (2011)*.

## H4S47C-anchored cleavage of budding yeast

Mutant and wildtype *S. cerevisiae* strains were grown to log phase at 30°C in YPD medium, harvested and used for labeling and cleavage reactions as described (*Brogaard et al., 2012a*). Briefly, the cell pellet was resuspended in spheroplasting buffer in 1M sorbitol, partially spheroplasted with lyticase, washed with 1M sorbitol 0.1% NP-40 and brought up in labeling buffer (1M sorbitol, 50 mM NaCl, 10 mM Tris–HCl pH 7.5, 5 mM MgCl$_2$, 0.5 mM spermidine, 0.15 mM spermine, 0.1% NP-40 and 0.1 mM EDTA). Labeling was performed by addition of 7 mM OP reagent in DMSO to 20% volume and incubation for 2 hr at room temperature followed by overnight incubation at 4 °C. Cell pellets were washed with sorbitol/NP-40, incubated with CuCl$_2$, washed with a sorbitol/NP-40-containing buffer, subjected to oxidative cleavage at room temperature for 20 min using 6 mM H$_2$O$_2$ and 6 mM 2-methylpropionic acid, and the reaction was quenched by addition of neocuproine to 0.28 mM. We varied the protocol in several ways, yet obtained comparable results (*Figure 2A*). In some experiments 15–30 min cleavage reactions were performed 3–5 times by resuspending the cell pellet in the original volume of mapping buffer, repeating addition of methylpropionic acid and hydrogen peroxide followed by re-centrifugation. To further reduce background, in some experiments we reduced the concentration of lyticase 1:10, the concentration of OP reagent 1:10, and/or increased the concentration of NP-40 twofold. In some experiments we also used a modified DNA extraction protocol (*Zentner et al., 2013*), repeating the RNAse A digestion, phenol-chloroform/chloroform extraction and ethanol precipitation to remove residual RNA before Illumina Tru-Seq paired-end library preparation as described (*Zentner et al., 2013*). In some experiments KAPA HiFi DNA polymerase (KAPABiosystems, Woburn, MA) was used in place of Phusion Polymerase.

AluI (New England Biolabs, Ipswitch, MA) cleavage reactions were performed according to the manufacturer's instructions on wildtype DNA either subjected to mock hydroxy radical cleavage or directly purified from cells using the Epicentre (Madison, WI) MasterPure Yeast DNA Purification Kit.

## Modified MNase-seq and native ChIP

Wild-type and H4S47C cells were cultured, harvested, lyticased and washed twice in 1M sorbitol 0.1% NP-40 as described for cleavage mapping (*Brogaard et al., 2012a*). Samples derived from 500 ml of cells at 2 × 10$^7$/ml were split in half, and one half of each was washed with 1M sorbitol 0.1% NP-40 for

OP labeling, while the other other half was suspended in 4 ml fresh MNase buffer (1 M sorbitol; 10 mM Tris-HCl, pH 7.5; 50 mM NaCl; 5 mM $MgCl_2$; 2 mM $CaCl_2$; 1 mM β-mercaptoethanol, 1 mM phenylmethanesulfonyl fluoride, + 1 protease inhibitor tablet [Roche, Nutley, NJ #04693159001] per 10 ml). Lightly lyticase-treated cells were digested with MNase for 10 min at 37°C using 4U MNase (Sigma–Aldrich, St. Louis, MO). OP labeling (2 hr at room temperature and overnight incubation at 4°C) was followed by three 1M sorbitol 0.1% NP-40 washes, then a sample of OP-treated cells was suspended in fresh MNase buffer and digested with MNase. MNase reactions were stopped by addition of EDTA to 10 mM and for MNase-seq, DNA was extracted.

For soluble and insoluble chromatin isolation and ChIP, the remaining samples were snap-frozen for storage, thawed on ice, and soluble chromatin was extracted and ChIP was performed as described (*Krassovsky et al., 2012*). An anti-Cse4 rabbit antibody was obtained from Sue Biggins and used at 5 µl per sample. Glycogen was added to ChIP samples before ethanol precipitation and Solexa library preparation. MNase fully penetrates into zymolyase-generated spheroplasts used for MNase-seq and ChIP, but hardly at all into the mildly lyticased cells used to allow penetration by OP-reagent for chemical cleavage. This resulted in >80% excess full-length DNA in the chromatin samples used for MNase-seq, which was removed prior to Solexa library preparation using a PCR cleanup kit (Clontech, Mountain View, CA). Illumina Tru-Seq libraries were prepared as described (*Zentner et al., 2013*).

## Data processing and analysis

Paired-end sequencing data were processed and aligned with Novoalign (Novocraft; http://www.novocraft.com) as described (*Henikoff et al., 2011*; *Krassovsky et al., 2012*) to Version 64 of the genome build from the *Saccharomyces* Genomic Database (SGD, http://www.yeastgenome.org), which corresponds to UC Santa Cruz SacCer3. Centromere CDEI-II-III coordinates are from SGD. Midpoint V-plots were constructed as described (*Krassovsky et al., 2012*). Fragments with midpoints within ± 1 kb of mid-centromeres were randomly sampled and plotted to equalize coverage to the least populated sample. For analysis of cleavage positions, separate V-plots of each left and right fragment end rather than the fragment mid-point were used. For H4S47C-anchored cleavage experiments two biological replicates (using KAPA DNA polymerase for library amplification) and one technical replicate (using Phusion DNA polymerase) gave virtually identical results and so fragment counts were combined for data analysis and presentation.

A profile of 15 centromeres (length 117–120 bp) comprising 111 base-pair positions was constructed by aligning at the left-most peak, excluding Cen4 (length 111 bp). The mean, standard deviation and minimum and maximum values of the centromeres at each position were computed. Alignments of the profile along the chromosome were filtered as follows: (1) The absolute value of z-score at each position was allowed to be ≥3 at no more than 3/111 positions. (2) The alignment was required to contain at least one position ≥ the smallest maximum of the profile, which was 186 (for Cen8). The Pearson correlation coefficient was computed for alignments passing the two filters. The same two filters were used for the delete-one jackknife, which in at least some cases resulted in exclusion of the deleted centromere prior to alignment.

## Watson and Crick strand analysis of cleavage sequencing data

W-C distributions were generated by determining for the left end of each fragment the number of fragments whose right ends were at a given distance. This analysis was performed for a distance range of −40 to +40 and normalized by total number of left-right combinations within the same distance range. W–W' distributions were generated by determining for the left end of each sequencing fragment the number of sequencing fragments whose left ends were at a distance in the range of 1–40 and normalized by the total number of left–left combinations within the same range. C–C' distributions were generated similarly for right ends.

## Gel-shift and AFM analysis of reconstituted particles

We reconstituted H3/H4 and Cse4/H4S47C (Cse4) octasomes with 147-bp duplexes of either the 601 positioning sequence or Cen3 following a 1 min room temperature trypsinization to minimize aggregation, as previously described (*Codomo et al., 2014*). After dialysis from 2M NaCl to 0.25 mM HEPES pH7.5 and 6% native PAGE, we excised the gel-shifted bands and extracted the nucleosomes as described (*Codomo et al., 2014*), except that extraction was done in OP labeling buffer rather than in water. An aliquot of each sample was incubated at 4°C for 15 hr with ¼ volume of OP-reagent to mimic the in vivo labeling procedure, and samples with or without treatment were resolved by 6% native PAGE.

For AFM, reconstituted H3/Cen3 and Cse4/Cen3 octasomes and Cse4/CDEII (78-bp) hemisomes in 0.25 mM HEPES were dialyzed vs OP-labeling buffer in a 5-step gradient, and one-half was incubated at 4°C for ~15 hr before gel purification. Bands were excised and particles were extracted into water and confirmed to be intact by native PAGE. Eluted samples were cross-linked in 0.6% glutaraldehyde for 25′ and imaged as described (*Codomo et al., 2014*).

## Data availability

Sequencing data generated in this publication have been deposited with GEO (GSE51949, *Henikoff et al., 2013*).

## Acknowledgements

Dedicated to the memory of Jonathan Widom who guided the initial stages of the project. We thank the FHCRC Genomics Shared Resource for Solexa sequencing, the FHCRC Electron Microscopy Shared Resource for AFM support, Sue Biggins for the Cse4 antibody, Takehito Furuyama for helpful discussions, and Paul Talbert, Jitendra Thakur, Anna Drinnenberg, Chris Weber and Florian Steiner for critical comments on the manuscript. This work was supported by the Howard Hughes Medical Institute.

## Additional information

### Funding

| Funder | Grant reference number | Author |
|---|---|---|
| Howard Hughes Medical Institute | | Steven Henikoff, Srinivas Ramachandran, Terri D Bryson |
| National Institutes of Health | U54 CA143862 | Christine A Codomo, Jorja G Henikoff |
| National Institutes of Health | U54 CA143869 | Kristin Brogaard, Jonathan Widom, Ji-Ping Wang |

The funders had no role in study design, data collection and interpretation, or the decision to submit the work for publication.

### Author contributions

SH, Conceived of the project, Performed the in vivo experiments, Performed the reconstitution experiments, Performed the analysis, Wrote the paper; SR, Performed the molecular dynamics simulations, Performed the analyses; KK, TDB, Performed the in vivo experiments; CAC, Performed the reconstitution experiments; KB, Advised on the experiments, Performed the reconstitution experiments, Contributed unpublished essential data or reagents; JW, Conceived of the project, Advised on the experiments, Contributed unpublished essential data or reagents; J-PW, Advised on the experiments, Contributed unpublished essential data or reagents; JGH, Performed the analyses

## Additional files

### Major dataset

The following dataset was generated:

| Author(s) | Year | Dataset title | Dataset ID and/or URL | Database, license, and accessibility information |
|---|---|---|---|---|
| Henikoff S, Ramachandran S, Krassovsky K, Bryson TD, Brogaard K, Widom J, Wang J, Henikoff JG | 2013 | The budding yeast Centromere DNA Element II wraps a stable Cse4 hemisome in either orientation in vivo | GSE51949; http://www.ncbi. nlm.nih.gov/geo/query/acc. cgi?acc=GSE51949 | Publicly available at the Gene Expression Omnibus (http://www. ncbi.nlm.nih.gov/geo/). |

**Reporting standards:** Standard used to collect data NCBI SRA and GEO protocols

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
