## [Decision Letter]

Thank you for sending your work entitled “The budding yeast Centromere DNA Element II wraps a stable Cse4 hemisome in either orientation in vivo” for consideration at *eLife*. Your article has been evaluated by a Senior editor and 2 reviewers, one of whom is a member of our Board of Reviewing Editors.

Both reviewers believe that the paper addresses an important question and is of sufficient interest for publication in *eLife*. However, one reviewer has noted, and the other reviewer concurs upon discussion, that significant problems need to be addressed experimentally. The general problem relates to the method for mapping the position of H4 on centromeric nucleosomes and the need for controls to deal with these potential problems. Given the controversy about the structure of centromeric nucleosomes, it is important to rule out artifactual explanations for the observed results (which themselves are not in question). Control experiments that are needed are listed below in the order of importance. Experiments 1-3 are viewed as essential, whereas 4-5 are suggested but not required.

1) Genome–wide MNase map of H4S47C cells under conditions used for the Widom approach to show that the long in vitro incubation doesn't affect MNase map at centromeres and other loci (i.e., an independent method to show that nucleosome positioning at centromeres in unaffected by conditions used for the cleavage approach).

2) Assay of wt yeast strain by the Widom approach to show that DNA cleavages at centromere depend on the modified histone.

3) Chemical cleavage map for reconstituted Cse4 octasomes and hemisomes.

4) ChIP of CBF1 and CBF3 in H4S47C before and after treatment of used for the chemical labeling as another assay of centromere function under the conditions used.

5) Assay of yeast strain to ensure that centromere function is normal in the H4S47C strain (e.g., frequency of chromosome loss or some other assay).

[Editors' note: further clarifications were requested prior to acceptance, as described below.]

We recognize that the revised manuscript is significantly improved, and it is not the usual *eLife* policy to have a second round of revision. However, critical experiment 3 wasn't done. This experiment was required because of a potential artifact (octasome to hemisome conversion during the long incubation period) that might explain the data in a manner different from the authors' conclusions. One can argue about how likely this artifact is, but it is not ad hoc; i.e., centromeric octasomes are cold-sensitive, and the critical assay involves long incubation in the cold.

We believe that the experiment is required to convincingly prove the central conclusion of the paper, although we would be very interested in the authors' opinion on this issue (perhaps they have some reason for excluding the artifactual possibility). We would be prepared to accept a modified manuscript that softens the conclusions – say by “evidence for” or “suggest” before the declarative statement in the title and related softened statements in the abstract. The current title and abstract is too definitive and must be changed. The authors must discuss the potential artifact as a possibility, and in general cannot claim that they have definitely established the conclusion in the title of the paper.

The authors do have good evidence, and the subject matter is important and interesting, which is why we are willing to accept a modified paper. However, the authors should either 1) present a modified paper with no new experiments as described or 2) present essentially the same paper but with experiment 3 performed.

Reviewer 2's comments are below for your consideration as you prepare to resubmit.

Reviewer 2:

The authors have clearly and adequately addressed essential experiments 1 and 2, but with one important exception, 3. The authors also provided important data for non-essential experiments 4, 5.

Essential experiment 3 (Chemical cleavage map for reconstituted Cse4 octasomes and hemisomes) remains incomplete. Authors state 'obtaining cleavage maps of Cse4 octamers on centromere-specific DNA is likely to be unfeasible,' citing Camahort et al. Mol Cell 35:794-805 “All attempts to reconstitute Cse4 nucleosomes using a centromeric DNA sequence and salt dialysis failed (data not shown).” However, the authors do not cite a subsequent publication, Xiao et al. Mol Cell 43:369-80 showing successful reconstitution under mild conditions: “...examined the temperature dependence of Cse4 nucleosome reconstitution on CEN3 and CEN4 DNA and found substantial increases in reconstitution efficiency of an octameric Cse4 nucleosome at temperatures of 23**°**C and 37**°**C (Figures S7A and S7B).” Furthermore, three independent reports demonstrated an alternate means of reconstituting octameric Cse4 nucleosomes, without salt dialysis, by use of the Scm3 centromeric histone chaperone (Dechassa et al. Nat Commun 2: 313; Shivaraju et al. J Biol Chem 286: 12016-23; Xiao et al. Mol Cell 43: 369-80. Therefore, there are robust, established procedures to reconstitute Cse4 octasomes.

It is crucial to examine the chemical cleavage map of such reconstituted Cse4 octasomes. Although authors' revised paper shows that chemical cleavage conditions do not change the MNase cleavage pattern of the Cse4 nucleosome in native chromatin – establishing retention of Cse4 under chemical cleavage conditions – the MNase map does not distinguish between Cse4 octasome and Cse4 hemisome. Without requested data, it is impossible to exclude the possibility that the distinctive chemical cleavage pattern at centromeres is an artifact of Cse4 octasome dissociation to hemisome under chemical cleavage conditions. That is why this control experiment is one of the essential three requested by reviewers. The known cold instability of Cse4 octasomes, as noted by authors, only underscores the importance of this control.

---

## [Author Response]

*Control experiments that are needed are listed below in the order of importance. Experiments 1-3 are viewed as essential, whereas 4-5 are suggested but not required*.

We have addressed each of these five points with additional experimental evidence, and the revision includes new supplementary figures and text revisions as needed to address each point. In each case, the additional experiments and analyses fully support our conclusions, as detailed below.

*1) Genome–wide MNase map of H4S47C cells under conditions used for the Widom approach to show that the long* in vitro *incubation doesn't affect MNase map at centromeres and other loci (i.e., an independent method to show that nucleosome positioning at centromeres in unaffected by conditions used for the cleavage approach)*.

We have performed MNase mapping on both H4S47C and wild-type cells in parallel both before and after the long labeling reaction. To compare nucleosome positioning over the centromere, we use fragment midpoint V-plot analysis, which allows precise particle positioning to be determined independent of particle occupancy (Henikoff et al. PNAS 2011, PMID:22025700). The four V-plots are nearly identical, with a sharp vertex over the middle of the 16 aligned centromeres indicating protection of ∼125 bp (last paragraph of the Results section entitled “Oppositely oriented hemisomes are present on different sequences…” and new Figure 6–figure supplement 3). We confirmed the MNase-seq data with V-plot of protected fragments by profiling insoluble chromatin (new Figure 6–figure supplement 4) which is enriched ∼100-fold for the kinetochore complex (Krassovsky et al., PNAS 2012). This centromere enrichment provides sufficient sequencing depth for near-base-pair resolution, and shows precise positioning and protection of the functional centromere. Similar midpoint V-plot patterns were obtained by native Cse4 ChIP-seq as described below in response to Point 4. We conclude that the H4S47C mutation and the long in vitro incubation have no effect on Cse4 positioning at the centromere.

*2) Assay of wt yeast strain by the Widom approach to show that DNA cleavages at centromere depend on the modified histone*.

This is an excellent suggestion. The cleavage frequency in a wt strain is too low to obtain a library comparable to that of the H4S47C strain, which requires that two cleavages are sufficiently close to generate short fragments. Therefore, we used digestion with the 4-cutter AluI prior to Solexa library preparation to generate fragments for paired-end sequencing. Performing the labeling and cleavage reaction protocol on a wild-type strain resulted in very few cleavages over centromeres relative to regions on either side (penultimate paragraph of the Results section entitled “Oppositely oriented hemisomes are present on different sequences…” and new Figure 6—figure supplement 2), in contrast to the sharply defined cleavage patterns observed over CDEII in the H4S47C strain (Figure 6). This result demonstrates that the DNA cleavages that we have mapped are indeed H4S47C-specific, and are not the result of OP-labeling of other cysteines on DNA-binding proteins around the kinetochore.

As an additional control, we identified background cleavages by isolating DNA from untreated cells and digesting with AluI before library preparation. This revealed that the level of background cleavage over centromeres is similarly low regardless of whether or not cells were subjected to OP-labeling and cleavage reactions, as expected if centromeres are fully occupied and protected during the cleavage reaction (penultimate paragraph of the Results section entitled “Oppositely oriented hemisomes are present on different sequences…” and new Figure 6—figure supplement 2).

*3) Chemical cleavage map for reconstituted Cse4 octasomes and hemisomes*.

The Richmond in vitro cleavage experiment referred to by Reviewer 2 (Flaus et al. PNAS 93:1370-75, 1996) does not really apply to our OP-Cu^+^ experiments, as those authors derivatized H4S47C with a different compound, S-(2-nitrophenylsulfenyl)cysteaminyl EDTA, an iron chelator. Therefore, we now include sequencing data for canonical OP-labeled octasomes reconstituted onto 601 DNA (last paragraph of the Results section entitled “A molecular dynamics model for H4S47C-anchored hydroxyl radical cleavage” and new Figure 1—figure supplement 1). The major cleavage peaks that we see for OP-Cu^+^-labeled 601 octasomes in vitro are among those predicted by our structural model and observed for octasomes in vivo. As the position of H4S47C relative to the surrounding residues and DNA is identical within experimental error when the H3 (1KX5) and Cse4 (3AN2) crystal structures are aligned, undoubtedly the same results would be obtained with Cse4 in place of H3 using 601 or other strongly positioning DNA. However, Cse4 octasomes reconstituted using native yeast centromeric DNA are unstable, presumably owing to the extreme AT-richness of CDEII. For example, a paper from J. Gerton, J. Workman and A. Shilatifard (Camahort et al. Mol Cell 35:794-805, 2009 PMID:19782029) stated: “All attempts to reconstitute Cse4 nucleosomes using a centromeric DNA sequence and salt dialysis failed (data not shown).” Although K. Luger’s group was successful in reconstituting Cse4 nucleosomes on 147-bp Cen3 DNA, they found that storage at 4 ^o^C overnight caused them to dissociate (Dechassa et al. Nat. Communications 2:313, 2011 PMID: 21587230). As these are similar to the conditions that we use for the OP-labeling reaction, obtaining cleavage maps of Cse4 octamers on centromere-specific DNA is likely to be unfeasible.

More relevant to the issue at hand is the in vitro cleavage pattern for hemisomes assembled on CDEII DNA. To address this question, we have taken advantage of our recent success in assembling hemisomes on short DNA duplexes. We follow a standard salt dialysis protocol but the DNAs that we use are only 62-78 bp long. During dialysis from 2 M NaCl to low salt or 4 M urea, “pseudo-octasomes” split to form two hemisomes (Furuyama et al. NAR 41:5769-83, 2013 PMID: 23620291). To obtain homogeneous hemisomes necessary for this experiment, we have found that trypsinizing octameric cores to partially remove the tails prevents aggregation that otherwise occurs, and elution of the resulting gel-shifted band from a gel suffices to recover pure hemisomes as judged by re-electrophoresis and AFM (Codomo et al. NSMB, 21:4-5, 2014 PMID:24389542). Using this refined protocol, we have performed in vitro cleavage mapping on gel-purified hemisomes where one of the oligos in the duplex was labeled on the 5’ end with Cy3 and on the 3’ end with Alexa488, using the gelFRET signal between them when wrapped around the particle to confirm that the gel-shifted band consists of wrapped hemisomes (no FRET is observed for the unshifted control band – Codomo et al, op cit). We electrophoresed the reaction products on sequencing gels alongside Maxam-Gilbert ladders using the Cy3 and Alexa488 fluorophores as end labels, which allowed us to determine the migration of both fragments generated by cleavage in the same gel lane. We find that the strongest cleavage peaks approximately correspond to the left-hand preferred cleavage sites seen for Cen4 in vivo (Materials and Methods section “In vitro cleavage mapping” new Figure 4—figure supplement 1). This direct approach determines the positions of single-strand cleavages measured from both ends of the same strand, as opposed to recovery of only double-strand cleavage fragments after end-polishing and adapter ligation for NGS library preparation. As expected for the loss of a single base during the cleavage reaction, we see that there is one base position between the left fragment cleavage site and the estimated right fragment cleavage site, and we see this for both peaks on the left side of CDEII. This is the side where the strong pair of in vivo cleavages are seen for Cen4 (Figure 4—figure supplement 2). Although only one clear peak is seen on the right side of Cen4 CDEII, the in vivo peaks on this side are weak. The fact that we are seeing the expected single-strand cleavages in vitro on hemisomes assembled on CDEII DNA confirms the predictions of our structural model.

*4) ChIP of CBF1 and CBF3 in H4S47C before and after treatment of used for the chemical labeling as another assay of centromere function under the conditions used*.

Reviewer 2 was concerned specifically about the occupancy of CBF1 and CBF3 after the long OP-labeling incubation reaction. This is addressed by the MNase-seq V-plots (last paragraph of the Results section entitled “Oppositely oriented hemisomes are present on different sequences…” and new Figure 6–figure supplements 3-4), which show full occupancy of the entire functional centromere, including CDEI and CDEIII, respectively the binding sites for Cbf1 and Cbf3. To perform ChIP requires either a tag or an antibody, and we are concerned about using tagged proteins that might themselves destabilize the centromere, especially components of the essential CBF3 complex (Cbf1 is non-essential with only a mild segregation phenotype). However, we obtained a custom anti-rabbit Cse4 antibody from Sue Biggins that she had used for ordinary ChIP, which has allowed a direct comparison between the mutant and wild-type strains before and after the OP-labeling treatment without any tags. Using our previously published native ChIP procedure (“ORGANIC” profiling – Krassovsky et al. PNAS 2012 and Kasinathan et al. Nat. Methods, 2013 PMID: 24336359) we obtained sufficient high-quality paired-end reads using the Biggins antibody to confirm the Cse4-associated occupancy of CDEI, CDEII and CDEIII implied by MNase-seq profiling. We observed no difference in Cse4 particle positioning regardless of whether or not nuclei were subjected to the OP-labeling treatment or whether the strain was mutant or wild-type (new Figure 6–figure supplement 5). We conclude that neither the mutant nor the labeling treatment has a detectable effect on occupancy or positioning of centromere proteins.

*5) Assay of yeast strain to ensure that centromere function is normal in the H4S47C strain (e.g., frequency of chromosome loss or some other assay)*.

The concern raised by Reviewer 2 was based on the following statement in Brogaard et al. (Meth Enz 2012): “The H4S47C/ura3 strain is viable but maintains a growth phenotype and is temperature sensitive”. In response to the reasonable concerns raised by Reviewer 2, we have now more thoroughly investigated phenotypic differences between the H4S47C and its parent BY4741 strain. Most relevant to our study and that of Brogaard et al. (Nature 2012), we find that the H4S47C and BY4741 log growth rates are virtually superimposable with a 90 minute doubling time, and we now provide growth and viability measurements for the two strains (new Figure 2—figure supplement 1). Nevertheless, consistent differences are seen when cells are grown on plates (mutant colonies are smaller), when cells are inoculated into fresh culture medium (mutant cells take longer to recover), and when cells reach stationary phase (mutant cells grow to a higher density). Furthermore, mutant cells are ∼8 % larger in diameter than wild-type cells, which suggests a failure of the strain to achieve a quiescent state upon nutrient deprivation (Linda Breeden, personal communication). For example, the uppermost cells in a colony become increasingly nutrient-deprived as the colony grows, limiting colony growth. Also, cells in liquid culture that fail to enter a quiescent state will not halt protein synthesis and stop dividing in time to prevent metabolic damage. To test this explanation, we extended the culture period, and found that the H4S47C strain showed a dramatic loss of viability (from ∼95 % to ∼60 %) about one day after reaching stationary phase, whereas the parent strain showed only a minor loss of viability (from ∼95 % to ∼90 %). This confirms that the phenotypes result from failure to enter a quiescent state upon nutrient deprivation. We also tested for temperature sensitivity in suspension, but found that the mutant cells if anything grew better than wild-type cells at 37^o^C in liquid culture (new Figure 2—figure supplement 1).

A parsimonious explanation for the failure to enter a quiescent state is reduced H4 dosage. Each *S. cerevisiae* histone is present in two copies encoding the same protein, and only one of the two H4 genes (HHF1 to Cys47) and was mutated while the other (HHF2) was replaced with a selectable marker (URA3). This is a routine strategy for mutating histones; for example, the Boeke lab replaced both HHF2 and HTF2 (histone H3 gene 1) to construct 486 single mutations in H3 and H4 (Dai et al. Cell 2008, PMID: 18805098), and as a result, a similar syndrome to what we are observing for H4S47C is a feature of histone mutations that is well-known to yeast chromatin researchers (S. Biggins, personal communication). HHF2 deletions, but not HHF1 deletions, cause reduced starvation resistance (Davey et al. Environ Microbiol 2012, PMID:22356628), which is consistent with the HHF2 deletion in the H4S47C mutant strain causing the phenotypic syndrome that we observe in this strain. Moreover reduced histone levels are associated with reduced life-span in yeast (Feser et al. Mol Cell, 2010, PMID:20832724). As our experiments were performed during early–mid log-phase growth, these phenotypes are not relevant to our study.

We also performed plasmid loss assays in the H4S47C strain. Following the standard protocol, we grew cultures without selection for 7–11 generations, then measured plasmid loss. We found that a centromere-containing single-copy plasmid is retained in the mutant strain in 99 % of the segregations over this period of growth, whereas the parent strain (with wild type HHF1 and HHF2) showed 99.99% retention (new Figure 1–figure supplement 2C legend). Given that these plasmid loss measurements were made after cells had approached stationary phase, it is likely that the 1% plasmid loss in the mutant strain is related to the syndrome described above, not reduced centromere function. In any case, the demonstration that 99% of CEN plasmid segregations occur normally validates our conclusions.

[Editors' note: further clarifications were requested prior to acceptance, as described below.]

*We recognize that the revised manuscript is significantly improved, and it not the usual eLife policy to have a second round of revision. However, critical experiment 3 wasn't done. This experiment was required because of a potential artifact (octasome to hemisome conversion during the long incubation period) that might explain the data in a manner different from the authors' conclusions. One can argue about how likely this artifact is, but it is not* ad hoc*; i.e., centromeric octasomes are cold-sensitive, and the critical assay involves long incubation in the cold*.

We are pleased that the Editors and Reviewer 2 have accepted nearly all of our essential and non-essential experiments, which include numbers 1, 2, most of 3, 4 and 5. However, the above assessment of our rebuttal is not accurate. In the first round of review, Reviewer 2 asked that we provide in vitro cleavage mapping data for the following reason:

“What is the in vitro chemical cleavage map for reconstituted canonical nucleosomes, Cse4 octasomes and hemisomes under DMSO-OP conditions? Instead of molecular simulations, determining the actual sites of cleavage in vitro (as was done by Richmond for the canonical nucleosome) is necessary for interpreting in vivo data.”

In our understanding of this comment, the key cleavage data required was for Cse4 hemisomes on CDEII DNA, because that is what we see in vivo. We also provided cleavage data for H3 octasomes, because the Richmond study used a different cleavage reagent. We saw no reason to provide Cse4 octasome cleavage data in addition to H3 octasome cleavage data because the crystal structures precisely superimpose around H4S47, and so the cleavage maps should be identical. In other words, we performed the experiments required to address the specific issue raised, we described these experiments in the rebuttal and revision, and we included them as new figure supplements.

In the latest response, Reviewer 2 gives an entirely different reason for requiring in vitro data, namely to rule out an improbable scenario that we frankly had not previously considered:

“Without requested data, it is impossible to exclude the possibility that the distinctive chemical cleavage pattern at centromeres is an artifact of Cse4 octasome dissociation to hemisome under chemical cleavage conditions. That is why this control experiment is one of the essential three requested by reviewers. The known cold instability of Cse4 octasomes, as noted by authors, only underscores the importance of this control.”

This reason for performing the in vitro control experiment was not mentioned in the first review round, and in any case, we think it unlikely that an octasome-to-hemisome transition can occur without causing any detectable change in the position of the particles at the centromere in vivo. Our near-base pair resolution MNase and ChIP mapping showed that the position of the particles at the centromere is the same both before and after treatment and for both mutant and wild-type cells. Reviewer 2 is now arguing for a complete switch from a (right-handed?) octasome to a hemisome without any detectable change in positioning of the particles at the centromere.

As detailed below, the requested experiment 3 does not address the issue, However, we have now done a set of in vitro experiments that does. As a result, we have ruled out transitions from octasomes to any other forms, including hemisomes, hexasomes or tetrasomes. Our revision includes the new data (new Figure 7).

*We believe that the experiment is required to convincingly prove the central conclusion of the paper, although we would be very interested in the authors' opinion on this issue (perhaps they have some reason for excluding the artifactual possibility). We would be prepared to accept a modified manuscript that softens the conclusions* – *say by “evidence for” or “suggest” before the declarative statement in the title and related softened statements in the abstract. The current title and abstract is too definitive and must be changed. The authors must discuss the potential artifact as a possibility, and in general cannot claim that they have definitely established the conclusion in the title of the paper*.

*The authors do have good evidence, and the subject matter is important and interesting, which is why we are willing to accept a modified paper. However, the authors should either 1) present a modified paper with no new experiments as described or 2) present essentially the same paper but with experiment 3 performed*.

Experiment 3 does not address the possibility of an artifactual transition during the OP-labeling step for the following reasons:

1) To produce hemisomes in vitro, pseudo-octasomes assembled on 62-78 bp duplexes are split by dialysis from high salt to low salt or 4 M urea (Furuyama et al. (2013) *NAR*). We and others have also produced hemisomes in vitro by assembling on supercoiled plasmids in the presence of TopoI (Furuyama et al. (2009) *Cell*; Huang et al. (2011) *PNAS*; Shiveraju et al. (2012) *Cell*). However, nobody to our knowledge has ever produced stable hemisomes on 147-bp linear DNA, presumably because octasomes will outcompete hemisomes during reconstitution owing to having twice as many protein–DNA contacts and having protein–protein docking contacts between nucleosome halves. When these contacts are reduced, for example in 4 M urea + 300 mM NaCl, H2A/H2B dimers are released, leaving an (H3/H4)_2_ tetrasome (Yamasu & Senshu (1990) *J. Biochem*), as expected because the H2B/H4 4-helix bundles are weaker than the H3/H3 4-helix bundle. So although an octasome-to-hemisome transition might well occur in vivo, as has been proposed by others, it cannot be made to happen on 147-bp DNA. Even if it did, we would not expect to recover the products, much less map their cleavage patterns. It follows that cleavage mapping on 147-bp DNA can only result in an octasome pattern, and so this experiment will be uninformative with respect to a presumptive artifactual hemisome-to-octasome transition.

2) In vitro cleavage mapping of Cse4 octasomes would not detect an artifactual transition from a Cse4 octasome to a tetrasome, the form favored by Carl Wu (Xiao et al (2011) *Mol Cell* PMID:21816344), Steven Harrison (Cho & Harrison (2011) *PNAS* PMID:21606327) and Ajit Joglekar (Aravamudhan et al. *Curr Biol* (2013) PMID:23623551), as tetrasomes and octasomes should have identical positioning of H4.

3) It is not correct that “centromeric octasomes are cold-sensitive”. As Reviewer 2 pointed out, stable Cse4/Cen nucleosomes can be readily produced by salt dialysis, for example at higher temperature. We now realize that the observation we referred to in the rebuttal suggesting cold sensitivity was only Cen3 DNA assembled with Cse4 + Xenopus histone partners that disassembled upon storage (Figure S7 of Dechassa et al. 2011, Nat Comm). In fact, as Reviewer 2 pointed out, Cen3 with yeast histones remained intact upon storage in that study and others. We thank Reviewer 2 for correcting our erroneous suggestion that producing Cse4/Cen octasomes is likely to be unfeasible.

4) The protocol for in vitro cleavage does not involve incubating octasomes with OP reagent (explained below), so even if we had performed cleavage mapping on Cse4 octasomes, this would not have addressed this potential artifact.

Fortunately, we can test the possibility of an octasome-to-hemisome transition with a simple and direct set of in vitro experiments. The objective is to confirm the MNase-mapping and ChIP in vivo experiments that we provided in the first round by showing that the long incubation needed for labeling does not disrupt a Cse4 octasome in vitro. To do this we cannot use the in vitro cleavage protocol that we used in and Figure 4—figure supplement 2, because there we had labeled the histone octamers before assembling octasomes or hemisomes. Rather, we need to first assemble octasomes, subject them to the OP labeling procedure and then show that the octasomes remain intact. We have now performed this test. We assembled H3/601, H3/Cen3, Cse4-H4S47C/601 and Cse4-H4S47C/Cen3 octasomes using lightly trypsinized cores to avoid aggregation that otherwise occurs in the 1 M sorbitol-containing OP labeling buffer (data not shown). We used our recently published gel-shift protocol (Codomo et al. (2014) *NSMB* PMID:24389542) both to extract the octasome band into labeling buffer and to assay for possible breakdown after DMSO-OP treatment. The results are shown in the new Figure 7 and referred to in the second paragraoh of the Results section entitled “Cse4 hemisomes are stable at centromeres”. In all cases, the gel shift after overnight OP incubation was virtually identical between treated and untreated samples, with no evidence of new bands appearing as a result of either gel purification in labeling buffer or incubation with DMSO-OP. We confirmed this conclusion directly on single particles using AFM imaging: both H3-H4/Cen3 and Cse4-H4S47C/Cen3 particles were of octameric height whether or not they were subjected to a long OP incubation, using a Cse4-H4S47C/CDEII hemisome control assembled in parallel, which also remained intact during the labeling reaction. These experiments provide direct in vitro support for our conclusion based on native MNase and ChIP mapping that the labeling treatment could not have caused a transition from octasomes to hemisomes, hexasomes or tetrasomes. To highlight the fact that both in vitro and in vivo results demonstrate the stability of particles to the OP-labeling treatment, the new Figure 7 also includes the native MNase-seq data (now Figure 7, previously Figure 6–figure supplement 3).